# Research on coupling coordination and influencing factors between Urban low-carbon economy efficiency and digital finance—Evidence from 100 cities in China's Yangtze River economic belt

**Fengge Yao**[1]☯**, Liqing Xue**[1]☯**, Jiayuan Liang**[1]☯*

School of Finance, Harbin University of Commerce, Harbin, China

☯ These authors contributed equally to this work.
* timljy1994@gmail.com

**Data Availability Statement:** All relevant data are within the paper and its Supporting information files. The minimum data set for our manuscript is

## Abstract

China is a large country with rapid economic expansion and high energy consumption, which implies that the country's overall carbon emissions are enormous. It is vital to increase urban low-carbon economy efficiency (ULEE) to achieve sustainable development of China's urban economy. Digital finance is a significant tool to boost ULEE by providing a convenient and effective funding channel for urban low-carbon economic transformation. Analyzing the coupled and coordinated relationship between ULEE and digital finance is of vital importance for the sustainable development of the urban economy. This paper selects panel data of 100 cities in China's Yangtze River Economic Belt (YEB) in 2011-2019, and analyzes the research methods such as the Global Malmquist-Luenberger index model, coupling coordination degree (CCD) model, standard deviation ellipse model, gray model, and geographic detector by The spatial and temporal distribution, dynamic evolution characteristics and influencing factors of the CCD between ULEE and digital finance are analyzed. The study shows that: (1) the CCD of ULEE and digital finance grows by 3.42% annually, reflecting the increasingly coordinated development of the two systems; (2) The CCD of ULEE and digital finance shows a distribution pattern of gradient increase from the upstream region of Yangtze River to the downstream region, meanwhile, the spatial center of gravity moves mainly in the midstream region; (3) The spatial center of gravity of CCD of ULEE and digital finance is expected to move 22.17 km to the southwest from 2019 to 2040; (4) In terms of influencing factors, the influence of informatization and industrial structure on the CCD increases over time, while the influence of factors such as population development, greening, transportation, and scientific research decreases over time. Finally, this paper proposes policy recommendations for improving the CCD of ULEE and digital finance based on the empirical results.

derived from the China Urban Statistical Yearbook, which is available to the general public. The link to the database is https://data.cnki.net/yearbook/Single/N2022040095.

**Funding:** This work was supported by the National Social Science Foundation of China (17BJY119) and the Harbin University of Commerce 2020 Postgraduate Innovation Research Funding Program (YJSCX2020-624HSD). The funders provided the cost of data collection and the software needed for the study.

**Competing interests:** The authors have declared that no competing interests exist.

## Introduction

It is well known that global warming due to the continued growth of $CO_2$ emissions has not only brought about glacial melting, sea-level rise, and reduction in species diversity [1]. It also increases the frequency of extreme weather events such as floods, droughts, hailstorms, tropical storms, and tornadoes [2]. China is one of the countries with the highest total carbon emissions in the world [3]. In 2016, China's carbon emissions were 1.97 times higher than those of the United States, accounting for 27.49% of the world's total emissions. As one of the most important urban economic zones in China, the YEB not only spans the eastern, central, and western regions of China but also plays a vital role in China's economic development with a population size of 600 million people and a dense urban distribution [4]. In 2018, China's YEB region contributed 44.1% of China's GDP [5], but also produced the most carbon emissions [6]. The reason for this is that the YEB gathers many key national manufacturing projects such as steel, petrochemicals, automobiles, and electromechanics [7], and these manufacturing projects consume a large amount of fossil energy, which is what causes the total $CO_2$ emissions of the YEB to be too large [8]. In 2013, for example, the carbon emissions of YEB accounted for about 44.6% of the national carbon emissions. Some scholars point out that developing a low-carbon economy is an efficient strategy to deal with high energy usage and carbon emissions [9]. Low carbon economics states that ULEE can be seen as an environmental efficiency that measures the ability of an urban economic development process to increase regional GDP with fewer factor inputs and produce fewer carbon emissions [10]. Therefore, improving the ULEE of YEB is an important way to alleviate environmental pressure in China.

With the fast growth of digital finance, the connection between ULEE and digital finance has become closer. On the one hand, digital finance contributes the growth of the low-carbon economy. The advancement of digital finance improves the efficiency of financing [11], which can provide sufficient financial support for low-carbon industries, help low-carbon industries develop rapidly, encourage the improvement of industrial structure, and effectively reduce carbon emissions, thus boosting the growth of ULEE. At the same time, digital finance can also help high-carbon emission enterprises upgrade their energy utilization technologies through financial support [12], thus realizing the growth of the urban low-carbon economy. In addition, digital finance is convenient and inclusive [13], and digital financial services can be completed through cell phones, personal computers, the Internet, or with reliable digital payment systems [14], which can reduce the carbon emissions generated by SMEs and individual consumers traveling offline to and from financial institutions to participate in financial services. On the other hand, the improvement of ULEE can foster the healthy growth of digital finance. The improvement of ULEE means that the relevant resource elements in the region are better allocated and applied [15], which can provide a better soft and hard environment for the digital finance industry's growth. The growth of the low-carbon economy accelerates the transformation of industrial structure and accelerates the development of high-tech industries with low-carbon row characteristics, which will provide a higher level of digital technology support for digital finance. In addition, the improvement of ULEE indicates that the urban atmosphere is improved, which can improve the attractiveness of the city to financial and high-tech talents [16], thus enabling the growth of local digital finance. In summary, analyzing the development relationship between ULEE and digital finance in the YEB region and the influencing factors is of non-negligible relevance to the low-carbon transition and sustainable development of Chinese cities.

Current literature mainly focuses on calculating and analyzing the efficiency of low carbon economy in each region [17], while there is a gap in the research on the coupled coordination relationship between ULEE and digital finance. Meanwhile, the existing literature lacks the

analysis of the future spatial pattern of CCD between ULEE and digital finance and the exploration of the influencing factors of CCD. In addition, the assessment indicator system of low carbon economy efficiency in the existing literature considers a single perspective and needs to be improved [18]. Therefore, to remedy these shortcomings, this paper focuses on addressing three issues regarding the developmental relationship between ULEE and digital finance in the Chinese YEB region:

- How has the ULEE in the YEB region of China changed over the past few years?

- How has the coupling and coordination relationship between ULEE and digital finance in China's YEB region changed over the past few years, and how will the spatial pattern change in the future?

- What are the factors affecting the coordinated development of ULEE and digital finance in the YEB region of China?

These questions are closely related to urban low-carbon transition as well as sustainable development. The most important contribution of this study is to provide the first reliable model for evaluating the level of coordinated development of ULEE and digital finance, using 100 cities in the YEB region of China as examples. Based on the SDE model, gray model, and geographic detector, the spatial and temporal evolution characteristics, future spatial patterns, and impact factors of the coupled and coordinated relationship between the ULEE system and digital financial system are analyzed for the first time. In addition, this study improves the evaluation framework of low carbon economy efficiency from the viewpoint of public finance and land usage. This research provides a new path to foster the coordinated development of low-carbon economy efficiency and digital finance in the YEB region, which helps the government to formulate environmental protection policies, accelerate the low-carbon urban transformation, and boost sustainable development.

For the rest of the paper, "Literature review" section describes the literature related to this paper. "Materials and methods" section elaborates on the construction of the indicator system, sample selection, data sources, and models used in this study. "Results" section is a presentation of the empirical results. "Discussion" section is a discussion of the empirical results, answering the three research questions posed in the introduction and describing the implications and limitations of this study. Conclusions and policy recommendations are presented in "Conclusion and countermeasures" section.

## Literature review

The reliability of the research data ensures the credibility of the research conclusions. Currently, the official regional carbon emission data published in China are very limited, and the regional carbon emission data are mainly obtained through measurement. According to the existing literature, regional carbon emission measurement is mainly based on two methods: production-based accounting and consumption-based accounting [19]. Many scholars have measured $CO_2$ emissions using production-side-based accounting and conducted related studies. Shan (2018) measured China's energy-related carbon emissions and industrial process-related carbon emissions from 1997–2015 from a production perspective regarding IPCC guidelines [20]. Wang (2020) measured carbon emissions in China and India using a production-based approach and compared their dynamic evolution and drivers [21]. Some scholars argued that measuring carbon emissions based on the production perspective would ignore the transfer of emissions due to import trade and create a "carbon leakage" phenomenon [22]. The consumption-based carbon emission measurement can effectively avoid the

underestimation and overestimation of carbon emissions caused by trade [23], and researchers have also conducted a series of studies on this measurement method. Bai (2021) measured consumption-based carbon emissions in the Beijing-Tianjin-Hebei region of China and analyzed the differences and drivers of emissions between cities from 2012 to 2015 [24]. Qian (2022) measured the carbon emissions of 47 cities from the Pearl River Basin in China using a consumption-based carbon emission estimation method and found that the carbon emissions of 47 cities accounted for 13.1% out of China's emissions, and there were large variations in carbon emissions between cities [25]. It is worth mentioning that, on the consumption side, the building industry has great potential and research value in reducing carbon emissions, which has been analyzed and discussed by several researchers. Sun (2022) used bibliometric methods to analyze and summarize 364 articles published from 1990 to 2021 on peak carbon and carbon neutrality in the building sector [26]. Zhang (2022) evaluated the carbon emission reduction and carbon emission reduction efficiency of commercial buildings in China and the U.S. at different scales and mapped the energy efficiency improvement paths of commercial buildings in both countries [27]. Li (2022) established an assessment framework of emission reduction intensity, quantity, and efficiency through carbon intensity decomposition and evaluated the carbon emission reduction changes of commercial buildings in 30 provinces in China from 2001 to 2016 [28]. Xiang (2021) measured the carbon emissions of commercial buildings in China and then estimated them using LASSO regression, optimized the nonlinear parameters using a whale optimization algorithm, and found that the peak emissions in the commercial building sector were 1264.81 Mt $CO_2$, and the main drivers of carbon emissions were population size and energy intensity [29]. Xiang (2022) developed a novel open-source tool PyLMDI based on the LMDI method and used it to analyze the carbon reduction potential of commercial buildings in China and the United States [30].

Digital finance refers to all comprehensive applications that rely on digital technology to innovate traditional financial products and service forms. Its essence is to empower traditional finance to solve the problem of high risk and high cost arising from information asymmetry by using modern information technology such as artificial intelligence, cloud computing, blockchain, and big data, to change the link of value delivery in the traditional financial model, and to provide richer financial products while reshaping the traditional financial system. With the rapid growth of the low carbon economy, an increasing number of scholars have started to concern about the relationship between the low-carbon economy and finance. Several scholars argued that finance has a supportive function for the growth of the low-carbon economy [31], and on the one hand, finance can control carbon emissions by the carbon transaction market under the market mechanism [32]. Wang (2019) analyzed the effectiveness of carbon emissions trading pilots in China from 2007–2017 based on a robust regression algorithm with M estimation and found that carbon emissions intensity in China was decreasing year by year and was lower in regions with higher economic levels, and carbon emissions trading pilots had a significant driving effect on reducing carbon emissions [33]. Guo (2021) investigated the function of carbon emissions trading policies on the financing of carbon emission reduction and carbon emissions in China, and the findings showed that carbon emissions trading policies can effectively promote the financing of carbon emission reduction and reduce carbon emissions in China, with more significant effects in the eastern and affluent regions, and the effects were persistent [34]. On the other hand, finance can build links with the real economy by providing financing for low-carbon projects [35], bank loans [36], etc., so that the carbon emissions of emitters can be effectively reduced. Paroussos (2019) used a large-scale application of the CGE model in the context of global GHG emission reduction to measure the macroeconomic impact of the investments required to decrease the GHG emissions generated by the Italian energy system by 76% compared to 1990 levels. From the results, it was shown that

low-cost financial resources and clean energy technologies with market share and rapid advances would benefit Italy in its transformation to a low-carbon economy [37]. Schumacher (2020) analyzed the role of sustainable finance in supporting Japanese transformation to a net-zero carbon, sustainability based economy and evaluated the contribution of policies in expanding investments in sustainable finance and low-carbon infrastructure. The results revealed that the Japanese financial sector needs to widen the integration of sustainable finance and ESG policies to scale up financing for a net-zero carbon economy in all asset types through its investment portfolio [38]. Sartzetakis (2020) analyzed the important role of green bonds for the transition to a low-carbon development approach on the basis of the intergenerational bur-den theory and the need for large long-term infrastructure development [39]. Sun (2021) built an analytical model for the association between green finance and carbon emissions using neu-ral network technology and executed simulation tests to verify the validity of the results, and the results showed that a significant association existed between green finance and carbon emissions [40]. Elheddad (2020) studied the effect of e-finance on carbon emissions by select-ing panel data for 29 OECD countries from 2007 to 2016, controlling for possible heterogene-ity between countries using fixed and random effects models, and testing robustness using instrumental variables estimation methods and panel quantile regressions, which showed that the development of e-finance mitigates carbon emissions in OECD countries and plays an important role in environmental protection [41]. In addition, the low-carbon economy has an influential function in financial development. The development of the low-carbon economy cannot be separated from the strengthening of low-emission infrastructure, the vigorous devel-opment of clean energy, and the improvement of industrial structure, and the implementation of these activities will indirectly stimulate the development of the financial sector [42]. Obvi-ously, digital finance, as an innovation of traditional forms of financial services and financial products relying on digital technology, has a complex relationship with the low-carbon econ-omy that both promotes and constrains each other.

CCD is a method to analyze the correlation relationship between multiple systems, which can effectively reflect whether the relationship between multiple systems is harmonious and well-matched, with an overall trend of coordinated development [43]. At present, some researchers have applied the CCD model to investigate the coupled coordination relationship between different industries and carbon emissions. Han (2018) analyzed the CCD between agricultural carbon emissions and the agricultural economy in 30 Chinese provinces from 1997 to 2015 and studied the potential drivers using the LDMI decomposition model [44]. Pan (2021) used the SBM-DEA model combined with the CCD model to measure the coupled coordination of carbon emissions, economic development, and regional innovation in tourism and analyzed the core influencing factors using the geographic detector [45]. In addition, some scholars have studied the coupled coordination relationship between the carbon emis-sion system and other systems by constructing a CCD model. Shen (2018) measured the CCD between socio-economic and carbon emissions using an improved CCD model by selecting data from 30 Chinese provinces [46]. Song (2018) measured the coupled coordination rela-tionship between carbon emissions and urbanization using the coordination degree model and CCD model, respectively, based on data from 30 Chinese provinces [47]. Chen (2020) studied the degree of coordination between carbon emissions and the ecological environment in China from 2009 to 2015 using the CCD model and used the log-mean divisor exponential decomposition method to determine the key factors affecting the degree of coordination [48]. Zhou (2020) analyzed the CCD between carbon emission efficiency and industrial structure improvement in each province of China and designed the coupling paths using a distributional dynamics framework [49].

The above describes the data acquisition, the relationship between finance and low-carbon economy, and the choice of research methods. A review of the literature reveals that there are two specific knowledge gaps regarding the developing relationship between ULEE and digital finance:

- The current low carbon economy efficiency assessment index system is constructed from a single perspective and needs to be improved.

For most studies on measuring low-carbon economy efficiency, the assessment framework is mainly constructed around the capital, labor, energy, GDP, and carbon emissions [50], ignoring the role of public finance and land use in the advance of the low-carbon economy. Public finance can effectively stimulate low-carbon innovation [51], which has an important role in the low-carbon transition of cities. Meanwhile, land use is the second-largest source of carbon emissions after fossil fuels [52]. Considering the importance of public finance and land use to the growth of the urban low-carbon economy, this paper includes them in the evaluation index system of ULEE.

- Till now, there is no suitable indicator to measure the change in the coupled and coordinated relationship between ULEE and digital finance.

Traditional finance has undergone a digital transition, and digital finance is a new manifestation of that development. Digital finance can provide users with more convenient and efficient financial services with the help of the Internet and digital technology, and significantly increase the efficiency of financing for individuals and enterprises [53]. The existing researches have mainly concentrated on analyzing the relationship between low-carbon economy and traditional finance [54], and there are gaps in the research on the relationship between ULEE and digital finance, especially the research on the coupling and coordination relationship between the two systems. The coupling and coordination relationship between ULEE and digital finance lacks theoretical mechanisms and suitable measurement indicators. In addition, the influencing factors and future spatial pattern prediction of the CCD between the two systems also need to be studied. The study of the CCD between ULEE and digital finance and the influencing factors will help the coordinated development of the low-carbon economy and digital finance, accelerate the low-carbon transformation of Chinese cities, promote the process of global CO2 emission reduction, and enhance sustainable development.

Accordingly, this study endeavors to narrow these gaps through the following efforts:

- Improved the assessment indicator system of ULEE from the perspective of public finance and land use.

This research proposes a new evaluation index system of ULEE from seven aspects: capital, labor, public finance, land, energy, GDP, and carbon emission, and measures the ULEE of 100 cities in China's YEB region using the GML index model. Meanwhile, this paper analyzed the changes of ULEE from the time perspective, analyzed the distribution differences of ULEE from the spatial perspective, and decomposed the ULEE index to explore the reasons for the changes in ULEE.

- Measured the CCD of ULEE and digital finance in 100 cities in the YEB region of China from 2012 to 2019, and analyzed its influencing factors.

This paper measures the coupled coordination relationship between ULEE and digital finance in the YEB region for the first time by combining ULEE and Peking University Digital Inclusive Finance Index using the CCD model and conducts a spatio-temporal evolution analysis with the SDE model to discuss the changes in the CCD of the two systems and the reasons.

Meanwhile, this paper also conducts a time series prediction of the parameters of SDE based on the gray model and analyzes for the first time the changes in the spatial distribution of the CCD of ULEE and digital finance in 2019–2040. In addition, to further promote the coordinated development of ULEE and digital finance and sustainable urban development, this paper detects for the first time the changes in the influence of informatization, industrial structure, population development, greening, transportation, and scientific research on CCD between ULEE and digital finance by using geographic detectors.

## Materials and methods

### Index system

**Indicator selection of ULEE system.** ULEE reflects the degree of comprehensive utilization of input factors and the comparative relationship between inputs and desired outputs and non-desired outputs in the process of urban low-carbon economy development, and is often used to measure the allocation between the capital, labor, land, financial support, energy and the economic benefits generated by each region in accomplishing the goal of low-carbon emissions. In this paper, based on the actual situation of low-carbon economy development of cities in China's YEB, eight indicators of 100 cities in China's YEB from 2011 to 2019 were chosen as original data on the premise that the indicators are available and representative, based on the previous researches. The indicators are explained as follows:

1. Investment in fixed assets. A general term for the amount of work involved in the construction and purchase of nonliquid assets and the costs associated with it, expressed in monetary terms, over a certain period [55].

2. Number of employees in urban units. It refers to the number of people employed in urban units as of 24:00 on the final day of the current period and receiving earnings [56].

3. Land area of built-up area. It refers to the area of land in the urban administrative region that has actually been developed and built in pieces, with basic municipal utilities and public facilities [57].

4. Total LPG supply. It refers to the total amount of all liquefied petroleum gas provided by LPG enterprises to urban users, including the amount of liquefied petroleum gas purchased externally and wasted during transportation [58].

5. Total supply of artificial gas and natural gas. It relates to the total amount of gas supplied by gas enterprises to urban users, including the amount of gas obtained externally and lost in the process of transportation.

6. Local general public budget expenditure. It refers to the sum of local fiscal expenditure related to general public services, public safety, and other livelihood-related expenditures.

7. Gross regional product. It denotes the ultimate outcome of productive activities of wholly resident units in a region at market prices over a specified period [59].

8. Total annual electricity consumption. It describes the overall annual electricity consumption of assorted industries and the annual electricity consumption of civic and rural residents.

In terms of indicator processing, this paper measured the capital stock data of 100 cities in the YEB during the study period based on fixed-asset investment data using the perpetual inventory method with 2011 as the base period and deflated them by referring to the processing methods of Huang (2021) [60]. Since the Chinese government does not publish data on the total energy consumption of each city, this paper takes LPG, artificial gas, and natural gas

as the main energy sources of each city and converts the units of their consumption into standard coal according to the corresponding conversion coefficients, and obtains the main energy consumption of each city in YEB separately according to the processing method of Pan (2021) [45]. In addition, since urban carbon emissions are mainly generated by direct energy consumption and electrical energy consumption, this paper accounts for urban carbon emissions in terms of carbon emissions from direct energy consumption and carbon emissions from electrical energy consumption. The carbon emissions from direct energy consumption can be measured by utilizing the relevant transformation factors offered by IPCC [61]. For the accounting of carbon emissions from electricity consumption, this paper draws on Glaeser (2010) [62], which used the baseline emission factors of six regional power grids in North, Northeast, East, Central, Northwest, and South China over the years and the total annual electricity consumption of cities to calculate the carbon emissions from electricity consumption in each city. The accounting formula for urban carbon emissions can be expressed as follows:

$$C_t = \sum_{j=1}^{n} E_{tj} \times CF_j \times CCF_j \times COF_j \times B + D_t \times f_{ti}, \tag{1}$$

In the above equation, $C_t$ denotes the carbon emission of the city in period $t$, $E_{tj}$ denotes the consumption of type $j$ of energy in the city in period $t$, $CF_j$ denotes the conversion factor of type $j$ of energy, the LPG conversion standard coal factor is 1.7143kgce/kg, artificial gas and natural gas conversion standard coal factor are 1.3300kgce/m$^3$. $CCF_j$ denotes the carbon content factor of type $j$ of energy, the carbon content factor is 0.5041 kg/kgce for LPG and 0.4484 kg/kgce for manufactured gas and natural gas. $COF_j$ denotes the carbon oxidation factor of type $j$ of energy, $B$ denotes the amount ratio of carbon dioxide to carbon molecules (44/12), the carbon oxidation rate is 0.98 for LPG and 0.99 for manufactured gas and natural gas. $D_t$ denotes the total annual electricity consumption of the city in period $t$ and $f_{ti}$ denotes the baseline carbon emission factor for period $t$ in region $i$ where the city's grid is located.

In this paper, the capital stock was selected as the capital input indicator, the number of people are employed in urban units as the labor indicator, the land area of built-up area as the land input indicator, the main energy consumption of the city as the energy input indicator, the local general public budget expenditure as the fiscal input indicator, the gross regional product as the desired output indicator, and the urban carbon emission as the undesired output indicator. The indicator system of ULEE was constructed, as shown in Table 1.

**Index selection of digital finance system.** The index that is currently applied to evaluate the development degree of digital finance in China is more comprehensive and widely recognized by the academic community is the Peking University Digital Inclusive Finance Index. This index is calculated by a joint team composed of Peking University's Digital Finance Research Center and Ant Financial Services Group [63], and the set of indices has been widely

**Table 1. Index system of ULEE system.**

| First-Class Index | Second-Class Index | Third-Class Index | Unit |
|---|---|---|---|
| Input indexes | Capital input | Capital stock | $10^9$ yuan |
| | Labor input | Number of employees in urban units | $10^4$ people |
| | Land input | land area of built-up area | km$^2$ |
| | Energy input | Major urban energy consumption | Tons of standard coal |
| | Financial input | Local general public budget expenditure | $10^4$ yuan |
| Desired output indicators | Economic output | Gross regional product | $10^4$ yuan |
| Undesired output indicators | Carbon emission output | Urban carbon emissions | $10^4$ tons |

used in several studies related to digital finance in China [64, 65]. In this paper, the Peking University Digital Inclusive Finance Index was selected to evaluate the development level of digital finance in China, which constructs the indicator system of the digital financial system from three dimensions, including the breadth of digital financial coverage, the depth of digital financial use and the degree of digitalization of inclusive finance, and the weights of indicators in the indicator system are determined by a combination of the coefficient of variation method and the hierarchical analysis method. The indicator selection and related weights of the digital financial system are shown in Table 2.

## Data collection

China's YEB covers 11 provinces and cities in China, spanning the east, middle and west regions of China by the Yangtze River waterway, covering 21.4% of China's land area and

**Table 2. Index system of digital finance system.**

| First-Class Index | Second-Class Index | Third-Class Index |
|---|---|---|
| Breadth of coverage (54.0%) | Account coverage | Number of Alipay accounts per 10,000 people |
| | | Proportion of Alipay tied card users |
| | | Average number of bank cards tied to each Alipay account |
| Depth of use (29.7%) | Payment business (4.3%) | Number of payments per capita |
| | | Amount paid per capita |
| | | Ratio of the number of users active 50 times or more per year to the number of users active 1 time or more per year |
| | Money fund business (6.4%) | Number of Yuebao purchases per capita |
| | | Amount of Yuebao purchased per capita |
| | | Number of Yuebao purchases per 10,000 Alipay users |
| | Loan business (10.0%) | Number of Internet consumer loans per 10,000 adult Alipay users |
| | | Number of loans per capita |
| | | Amount of loan per capita |
| | | Number of Internet micro and small business loans per 10,000 adult Alipay users |
| | | Average number of loans per household for micro and small operators |
| | | Average loan amount for small and micro operators |
| | Insurance business (16.0%) | Number of insured users per 10,000 Alipay users |
| | | Number of insurance strokes per capita |
| | | Amount of insurance per capita |
| | Investment business (25.0%) | Number of Alipay users per 10,000 people involved in Internet investment and wealth management |
| | | Number of investments per capita |
| | | Investment amount per capita |
| | Credit business (38.3%) | Number of calls per natural person credit |
| | | Number of users using credit-based services per 10,000 Alipay users |
| Degree of digitization (16.3%) | Mobility (9.5%) | Percentage of mobile payment transactions |
| | | Percentage of mobile payment amount |
| | Affordability (16.0%) | Average loan interest rate for small and micro operators |
| | | Average personal loan interest rate |
| | Credit (24.8%) | Percentage of the number of transactions paid through Checklater |
| | | Percentage of the amount paid through Checklater |
| | | Percentage of Zhima Credit's no-deposit transactions (compared to all cases requiring a deposit) |
| | | Percentage of Zhima Credit's no-deposit amount (compared to all cases requiring a deposit) |
| | Facilitation (49.7%) | Percentage of user QR code payments |
| | | Percentage of the amount of user QR code payment |

**Table 3. Index system of digital finance system.**

| Region | Province | Cities |
|---|---|---|
| Upper Yangtze | Sichuan | Bazhong, Chengdu, Dazhou, Deyang, Guangyuan, Leshan, Luzhou, Meishan,Mianyang, Nanchong, Neijiang, Panzhihua, Yibin, Ziyang, Zigong |
| | Yunnan | Baoshan, Kunming, Lijiang, Qujing, Yuxi, Zhaotong |
| | Guizhou | Anshun, Guiyang, Liupanshui, Zunyi |
| | | Chongqing(municipality) |
| Middle Yangtze | Hubei | Ezhou, Huanggang, Huangshi, Jingmen, Jingzhou, Shiyan, Suizhou, Wuhan,Xianning, Xiaogan, Yichang |
| | Hunan | Changde, Chenzhou, Hengyang, Huaihua, Loudi, Shaoyang, Xiangtan, Yiyang, Yongzhou, Yueyang, Zhangjiajie, Changsha, Zhuzhou |
| | Jiangxi | Fuzhou, Ganzhou, Ji'an, Jingdezhen, Jiujiang, Nanchang, Pingxiang, Shangrao, Xinyu, Yichun, Yingtan |
| | Anhui | Anqing, Bengbu, Bozhou, Chizhou, Chuzhou, Fuyang, Hefei, Huaibei, Huainan, Huangshan, Liuan, Tongling, Wuhu, Suzhou(city in Anhui), Xuancheng |
| Lower Yangtze | Jiangsu | Changzhou, Huaian, Lianyungang, Nanjing, Nantong, Suzhou(city in Jiangsu), Taizhou (city in Jiangsu), Wuxi, Suqian, Xuzhou, Yangzhou, Zhenjiang, Yancheng |
| | Zhejiang | Hangzhou, Huzhou, Jiaxing, Jinhua, Ningbo, Quzhou, Shaoxing, Taizhou(city in Zhejiang), Wenzhou, Zhoushan |
| | | Shanghai(municipality) |

gathering 42.8% of China's population, with a dense distribution of cities, which has an important position in China's economic development. However, the YEB covers several key national industrial projects with strong energy consumption and heavy emissions, and the large amount of industrial pollution and greenhouse gases generated by these industrial enterprises has caused obstacles to China's low-carbon development. In recent years, with the speedy growth of digital finance in China's YEB, the financial support of digital finance for low-carbon projects has gradually increased, and the impact of digital finance in the YEB for the construction of China's low-carbon economy has become more and more obvious. Therefore, in this paper, 100 cities in China's YEB were selected as the sample, and referring to Pan (2020) [5], the 100 cities in China's YEB were divided into three regions according to their locations: upstream, midstream, and downstream of the Yangtze River, as shown in Table 3.

The research data in this paper are panel data of 100 cities in China's YEB from 2011–2019, and the original data are obtained from the China City Statistical Yearbook. For some missing data, this paper has used linear interpolation to complete them.

## Methods

**Global Malmquist-Luenberger (GML) Index Model.** The GML index model can be used for dynamic efficiency measurement. The model was proposed by Oh (2010) based on the study of Chung et al (1997) [66, 67]. In this paper, the GML index model was applied to calculate the ULEE in the YEB. The model has significant advantages and wide applicability, using production input and output data of all periods of the sample to construct a common production frontier surface based on the consideration of non-desired output, as a way to measure the gap between technical efficiency and frontier for each period and each decision unit.

Suppose there are $n$ decision units, each of which uses $m$ inputs to obtain $k_1$ desired outputs and $k_2$ undesired outputs. The vector of input indicators is $x = (x_1, x_2 \cdots, x_m) \in R_m^+$, the vector of desired output indicators is $y = (y_1, y_2 \cdots, y_{k_1}) \in R_{k_1}^+$, and the vector of non-desired output indicators is $b = (b_1, b_2 \cdots, b_{k_2}) \in R_{k_2}^+$. The set of possible production for the current

production period is:

$$P^t(x^t) = \{(y^t, b^t) \mid put\ into\ x^t\ product(y^t, b^t), t = 1, 2, \cdots, T\}, \tag{2}$$

Then the global production possibility set can be expressed as:

$$P^G(x) = P^1(x^1) \cup P^2(x^2) \cup \cdots \cup P^T(x^T), \tag{3}$$

Therefore, in this paper, referring to Färe (1994) [68], the functional equation for the GML index and its decomposition index is expressed as:

$$GM_t^{t+1}(x^t, y^t, b^t, x^{t+1}, y^{t+1}, b^{t+1}) = \frac{1 + D^G(x^t, y^t, b^t)}{1 + D^G(x^{t+1}, y^{t+1}, b^{t+1})}$$

$$= \frac{1 + D_{VRS}^t(x^t, y^t, b^t)}{1 + D_{VRS}^{t+1}(x^{t+1}, y^{t+1}, b^{t+1})} \times \frac{\frac{1 + D_{CRS}^G(x^t, y^t, b^t)}{1 + D_{VRS}^G(x^t, y^t, b^t)}}{\frac{1 + D_{CRS}^G(x^{t+1}, y^{t+1}, b^{t+1})}{1 + D_{VRS}^G(x^{t+1}, y^{t+1}, b^{t+1})}} \times \frac{\frac{1 + D_{VRS}^G(x^t, y^t, b^t)}{1 + D_{VRS}^t(x^t, y^t, b^t)}}{\frac{1 + D_{VRS}^G(x^{t+1}, y^{t+1}, b^{t+1})}{1 + D_{VRS}^{t+1}(x^{t+1}, y^{t+1}, b^{t+1})}}, \tag{4}$$

$$= PEC_t^{t+1} \times SEC_t^{t+1} \times TC_t^{t+1}$$

In the above equation, $D^G(x, y, b) = max\{\beta \mid (y + \beta y, b - \beta b) \in PG(x)\}$ is the global directional distance function, and the subscripts *CRS* and *VRS* indicate that the assumptions of constant returns to scale and changing returns to scale are satisfied, respectively. $GML_t^{t+1}$ indicates the total factor productivity change in period *t+1* relative to period *t*, which can be further decomposed into pure technical efficiency change ($PEC_t^{t+1}$), scale efficiency change ($SEC_t^{t+1}$), and technical progress change ($TC_t^{t+1}$). When $GML_t^{t+1}$ is greater than 1, less than 1, and equal to 1, it denotes total factor productivity growth, decrease, and no change from period *t* to period *t+1*, respectively. When $PEC_t^{t+1}$, $SEC_t^{t+1}$ and $TC_t^{t+1}$ are greater than 1, less than 1 and equal to 1, they indicate the growth, decrease and no change of this decomposition indicator from period *t* to period *t+1*, respectively.

**Coupling Coordination Degree (CCD) model.** The coupling degree reflects the degree of influence of the interaction between two or more systems. In this paper, the coupling degree model of dual system interaction was derived by drawing on the capacity coupling coefficient model in physics to evaluate the transition process between the disorderly and orderly states of the ULEE system and the digital financial system in China's YEB, and to show the dynamic correlation between the constituent elements within the system. Coupling degree can be expressed as follows:

$$C = \left\{ \frac{f(u) \times g(e)}{[f(u) + g(e)]^2} \right\}^{\frac{1}{2}}, \tag{5}$$

In the above equation, *C* represents the coupling degree, and the size of this value ranges from 0 to 1. The larger the value of coupling degree represents the stronger the interaction between the systems. $f(u)$ and $g(e)$ denote the comprehensive evaluation value of the ULEE system and the digital financial system, respectively.

The coupling degree can only reveal the degree of interaction between systems. In this paper, the CCD model was introduced to further quantify the degree of coordination between systems in the development process for a more deep and comprehensive analysis. The

**Table 4. Types of CCD between ULEE and digital finance.**

| Imbalanced Recession | | Coordinated Development | |
|---|---|---|---|
| CCD | Classification | CCD | Classification |
| 0.00–0.09 | Extreme imbalance | 0.50–0.59 | Reluctant coordination |
| 0.10–0.19 | Severe imbalance | 0.60–0.69 | Primary coordination |
| 0.20–0.29 | Moderate imbalance | 0.70–0.79 | Intermediate coordination |
| 0.30–0.39 | Slight imbalance | 0.80–0.89 | Well coordination |
| 0.40–0.49 | Approaching imbalance | 0.90–1.00 | High coordination |

calculation formulae are:

$$D = \sqrt{C \times T}, \tag{6}$$

$$T = \alpha f(u) + \beta g(e), \tag{7}$$

In equations, $D$ represents the degree of coupling coordination, the value ranges from 0 to 1, the larger the value the higher the degree of coordinated development between the two systems. $T$ is the combined development index of the two systems. $\alpha$ and $\beta$ are coefficients to be determined, this study considers both equally important, therefore $\alpha = \beta = 0.5$. In this paper, the coupling coordination between ULEE system and digital financial system was classified into 10 categories, and the classification is shown in Table 4.

**Standard Deviation Ellipse (SDE) model.** The standard deviation ellipse model was proposed by Lefever (1926), and this spatial statistical method can be used to analyze the directional characteristics of economic factors in spatial distribution, mainly including changes in the center and changes in discrete trends [69]. The standard deviation ellipse provides a quantitative description of the distribution characteristics of economic attributes in space through parameters such as center, long axis, short axis, and azimuth, and the arithmetic expressions of the relevant parameters are:

$$\bar{X}_w = \frac{\sum_{i=1}^n w_i x_i}{\sum_{i=1}^n w_i}; \bar{Y}_w = \frac{\sum_{i=1}^n w_i y_i}{\sum_{i=1}^n w_i}, \tag{8}$$

$$tan\theta = \frac{(\sum_{i=1}^n) w_i^2 \tilde{x}_i^2 - \sum_{i=1}^n) w_i^2 \tilde{y}_i^2) + \sqrt{(\sum_{i=1}^n w_i^2 \tilde{x}_i^2 - \sum_{i=1}^n) w_i^2 \tilde{y}_i^2)^2 + 4\sum_{i=1}^n w_i^2 \tilde{x}_i^2 \tilde{y}_i^2}}{2\sum_{i=1}^n w_i^2 \tilde{x}_i \tilde{y}_i}, \tag{9}$$

$$\sigma_x = \sqrt{\frac{\sum_{i=1}^n (w_i \tilde{x}_i cos\theta - w_i \tilde{y}_i sin\theta)^2}{\sum_{i=1}^n w_i^2}}; \sigma_y = \sqrt{\frac{\sum_{i=1}^n (w_i \tilde{x}_i sin\theta - w_i \tilde{y}_i cos\theta)^2}{\sum_{i=1}^n w_i^2}}, \tag{10}$$

In the above equations, $(x_i, y_i)$ denotes the spatial location of the study object, $w_i$ denotes the weight, and $(\bar{x}_w, \bar{y}_w)$ denotes the weighted mean center (MC), which indicates the center of gravity of the element in spatial distribution. $\theta$ is the azimuth of the ellipse, indicating the main trend direction of the element in space (i.e., the angle of clockwise rotation in the due north direction to the long axis of the ellipse). $\tilde{x}_i$ and $\tilde{y}_i$ are the coordinate deviations of each study object zone to the mean center, respectively. $\sigma_x$ and $\sigma_y$ are the standard deviations along the x-axis and y-axis, respectively, indicating the dispersion of the elements in the x-axis and y-axis directions, respectively.

**GM (1, 1) model.** The GM (1, 1) model is a prediction model based on gray system theory, and this prediction method has no strict requirements on the classification, sample content, and probability distribution of the research data, which is suitable for precise prediction of research problems with fewer data and poor information. The basic idea of this method is to accumulate the original sequence to weaken its intrinsic randomness and create an accumulative sequence with better regularity, and based on the new sequence, establish the differential equation and fit the system regularity to calculate the gray parameter, to construct a prediction equation model suitable for the application. The computational expressions of the model are:

$$dX^{(1)}/dt + aX^{(1)} = b, \tag{11}$$

The above equation is the differential equation corresponding to the GM (1, 1) model. $X^{(1)}$ denotes the new sequence obtained by accumulating $n$ sequence values and $t$ denotes the $n$th sequence value, $a$ denotes the development gray value, and $b$ denotes endogenous control gray value.

The parameter vector $\tilde{a}$ is obtained by least squares estimation as:

$$\hat{a} = (B^{\top}B)^{-1}B^{\top}Y = \binom{a}{b}, \tag{12}$$

In the above equation, $B$ is the data matrix and $Y$ is the data vector.
The computational expression of the cumulative series prediction model is:

$$\hat{X}^{(1)}(k+1) = [X^{(0)}(1 - b/a)]e^{-at} + b/a, k = 1, 2, \cdots, n, \tag{13}$$

The prediction model of the original series can be obtained by restoring the cumulative series, and the computational expression is:

$$\hat{X}^{(0)}(k+1) = \hat{X}^{(1)}(k+1) - \hat{X}^{(1)}(k), k = 1, 2, \cdots, n, \tag{14}$$

**Geographic detector.** The geographic detector is a statistical analysis method to analyze spatially stratified heterogeneity and reveal the influence of driving forces behind it. The method is not only immune to the problem of multicollinearity, but also better avoids the problem of endogeneity caused by the mutual causality of independent and dependent variables, and it is now widely used in many fields such as public health, regional economy, and ecological environment. In this paper, the factor detector in the geographic detector was selected to explore the role of each influencing factor on the CCD of ULEE and digital finance in China's YEB at different times periods. The model is calculated as follows:

$$q_{D,U} = 1 - \frac{1}{n\sigma_U^2}\sum_{i=1}^{m}n_{D,i}\sigma_{U_{D,i}}^2, \tag{15}$$

In the above equation, $q_{D,U}$ denotes the influence of driver $D$ on CCD, and takes a value between 0 and 1. Larger values indicate that the driver has stronger explanatory power on CCD. $n$ denotes the number of samples in the study area, $m$ denotes the number of types of each influence factor, and $n_{D,i}$ denotes the number of samples with type $i$ in influence factor $D$. $\sigma_U^2$ and $\sigma_{U_{D,i}}^2$ denote the variance of CCD for all samples within the study area and the variance of CCD for samples with type $i$ within the study area, respectively.

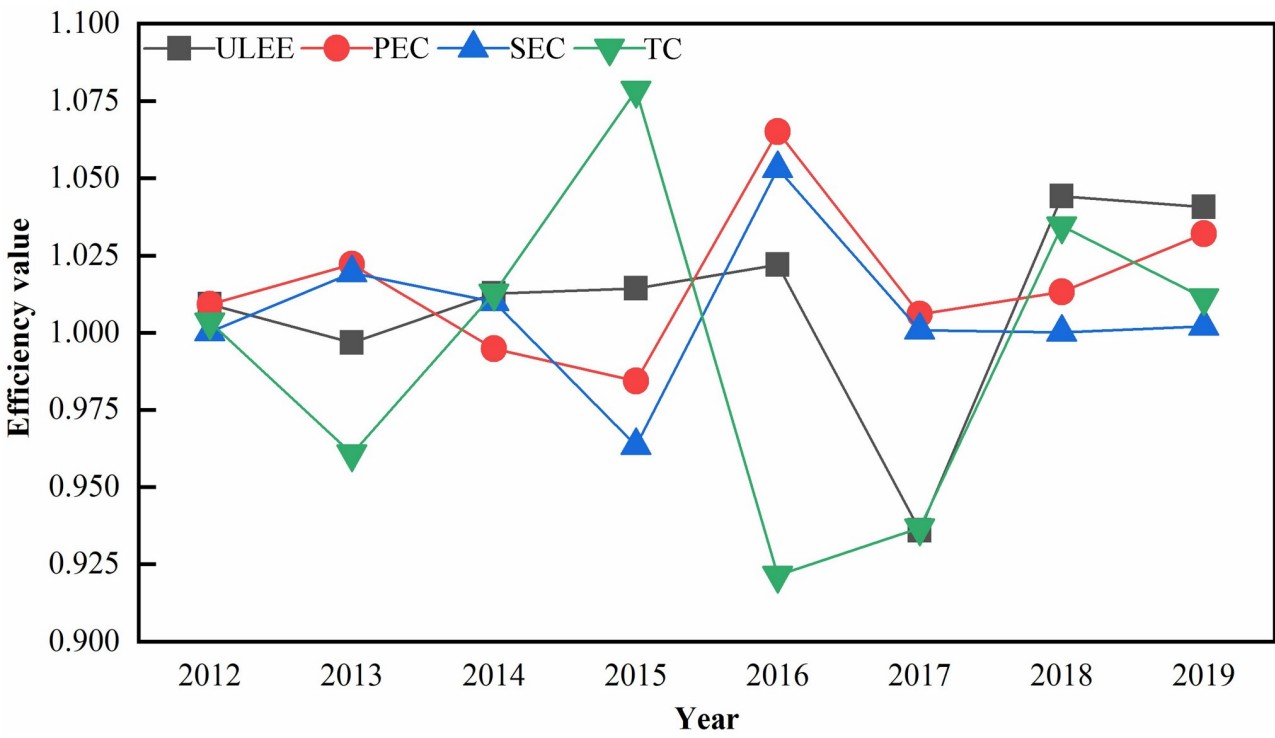

**Fig 1. Trend of the ULEE.**

## Results

### Analysis of ULEE

**Temporal characteristics of ULEE.** This paper used the GML index model to calculate the total factor productivity and decomposition index to measure the ULEE. The ULEE of the China YEB region from 2012 to 2019 is shown in Fig 1. From an overall perspective, the ULEE of China's YEB region shows a fluctuating upward trend with an average annual growth of 0.93% during the 8 years from 2012 to 2019. The years of ULEE growth amounted to 6 years, accounting for 75% of the entire 8 years. Meanwhile, ULEE increased by 7.41%, pure technical efficiency (PEC) increased by 13.16%, scale efficiency (SEC) increased by 4.80%, and technological progress (TC) decreased by 4.92% from 2012 to 2019. This indicates that the growth of PEC and SEC is the key cause for the growth of ULEE in China's YEB region from 2012 to 2019, with PEC contributing more to ULEE. Specifically, ULEE grew by 0.92% in 2012. A decomposition of ULEE showed that PEC, SEC, and TC increased by 0.92%, 0.03%, and 0.35% respectively, with PEC contributing the most to ULEE growth. ULEE experienced a small decline of 0.32% in 2013, while PEC and SEC increased by 2.21% and 1.94% respectively, and TC decreased by 3.93%. This indicates that the improvement in PEC and SEC failed to compensate for the negative impact of the decline in TC on ULEE, and the problem of backward carbon reduction technology in the YEB region is beginning to emerge, with energy-saving and emission reduction technology in urgent need of innovation. In 2014, ULEE showed an improvement compared to last year, with ULEE increasing by 1.27%, SEC and TC reaching the effective frontier, and only PEC with a value below 1. The ULEE still maintained its growth in 2015, with a growth rate of 1.4%. It is worth noting that TC in 2015 showed a significant increase with a growth rate of 7.84%, while PEC and SEC decreased by 1.56% and 3.66%

respectively, thus showing that the improvement of TC drove the growth of ULEE. The increase in ULEE in 2016 was the largest in eight years, with an increase of 2.2%. PEC and SCE both reached the effective frontier, increasing by 6.51% and 5.30%, respectively, while TC decreased by 7.85%, which shows that the growth of PEC and SEC drove the ULEE improvement. The decline in ULEE has increased significantly in 2017, decreasing by 6.40%, with ULEE at its lowest level in 8 years. Compared with 2016, TC in 2017 got slightly improved, but the growth of PEC and SEC decreased significantly and could not compensate for the loss caused by the decrease of TC. After 2017, ULEE started to show continuous growth, with 4.41% and 4.07% growth in 2018 and 2019, respectively. The decomposition of indicators shows that the growth rate of PEC is 1.33%, SEC is 0.01% and TC is 3.45% in 2018, which indicates that the rise of ULEE in 2018 is primarily due to the expansion of TC. While the growth rate of PEC in 2019 is 3.20%, SEC growth rate is 0.21% and TC growth rate is 1.13%, which means that PEC contributes the most to the growth of ULEE.

**Spatial differences in ULEE.** The differences in the spatial distribution of ULEE among cities in China's YEB are shown in Fig 2. From the overall view, the ULEE in the YEB region of China shows the spatial distribution of the strongest in the upstream region, the second strongest in the downstream region, and the weakest in the midstream region, and the average values of ULEE in the upstream, downstream, and midstream regions are 1.0172, 1.0077, and 1.0062, respectively. After decomposing the ULEE, it was found that the decomposition indexes showed spatial distributions with different characteristics in the YEB region, respectively. The spatial distribution of PEC is similar to that of ULEE, with the highest PEC level in the upstream region with a value of 1.0198 and the lowest PEC level in the midstream region with a value of 1.0118, while the PEC level in the downstream region is between the upstream and midstream regions with a value of 1.0155. The spatial distribution of SEC in the YEB region is mainly characterized by decreasing from the upstream region to the downstream region, with 1.0110, 1.0050, and 1.0030 for the upstream, middle, and downstream regions, respectively. The spatial distribution of TC in the YEB region is characterized by the strongest in the downstream zone, followed by the upstream zone, and the weakest in the midstream zone, with the values of TC in the downstream, upstream, and midstream regions being 0.9978, 0.9946, and 0.9934, respectively. From a city-specific perspective, 74 of the 100 cities in China's YEB region from 2012 to 2019 have average ULEE values above 1 and are at the production frontier. Among them, Yuxi, Jingzhou, Zunyi, Shanghai, Lijiang, Changzhou, Qujing, and Guangyuan had relatively high TFP, with mean values exceeding 1.03. While Zhaotong, Taizhou, Ziyang, Huzhou, Jiaxing, Dazhou, Changsha, Fuzhou, Nanchang, Yichang, Wenzhou, Liuan, Loudi, Jingmen, and other cities had relatively low efficiency of the low carbon economy, which was mainly affected by the low level of TC. Cities such as Xianning and Yichun had low PEC, which restricted the improvement of ULEE. The mean value of SEC in Ji'an and Jingmen is less than 1, which hinders the improvement of ULEE.

## Spatial and temporal evolution of the CCD

**Temporal evolution of the CCD.** In this paper, after dimensionless processing of the ULEE and digital inclusive finance indices of the China YEB region in 2012–2019, we obtained the ULEE composite index and digital finance composite index respectively, and calculated the CCD of the two systems from 2012 to 2019 according to the CCD model, and the changing trend of CCD is shown in Fig 3. From the perspective of the ULEE composite index, the ULEE of China's YEB showed an overall trend of increasing, then decreasing, and then increasing, with the index fluctuating between 0.2855 and 0.3583. This indicated that the ULEE in China's YEB was relatively stable, and there was still much potential for enhancement. From the

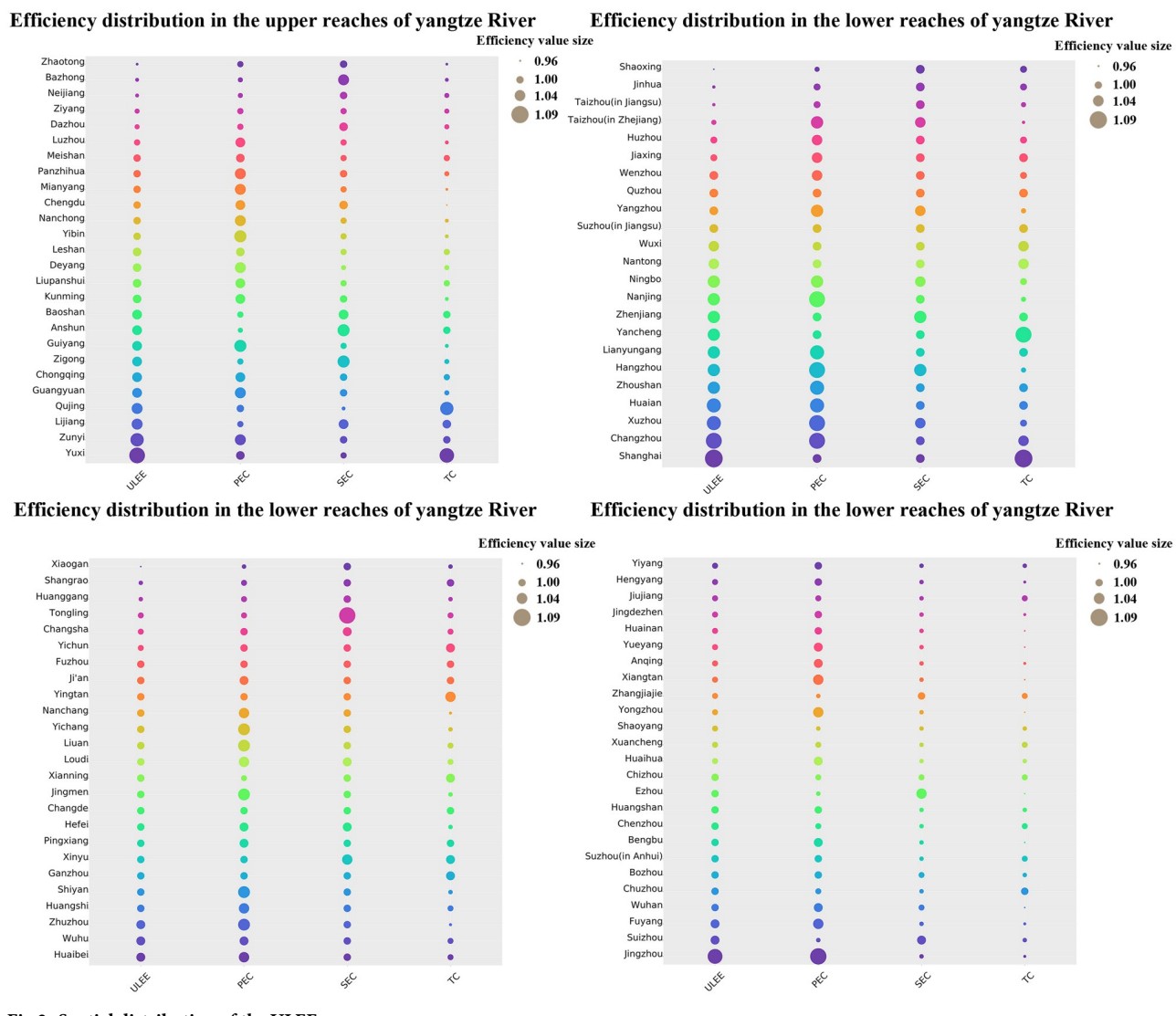

**Fig 2. Spatial distribution of the ULEE.**

perspective of the digital finance composite index, the level of digital finance development in China's YEB demonstrated a sustained rising trend, from 0.1276 in 2012 to 0.7315 in 2019, with an average annual increase of 59.16%, and the level of digital finance development in the YEB grew at the fastest rate from 2012 to 2013, and the growth rate declined year by year after 2015. This indicated that the growth of digital finance in China's YEB was good and has been at a high level, and there was still a trend to continue the development. From the perspective of CCD, the two systems of ULEE and digital finance in China's YEB showed the development trend of coupling optimization, and the CCD gradually changed from the approaching imbalance in 2012 to Intermediate coordination in 2019. From the perspective of CCD, the level of coordinated development of ULEE and digital finance in China's YEB regions generally shows an upward trend, with an average annual growth of 3.42% in CCD, and the CCD gradually transforms from the approaching imbalance in 2012 to Intermediate coordination in 2019. Specifically, the CCD maintained an annual increase from 0.44 to 0.65 during the period 2012–2016, but the value of the CCD is not high and the growth rate decreases year after year.

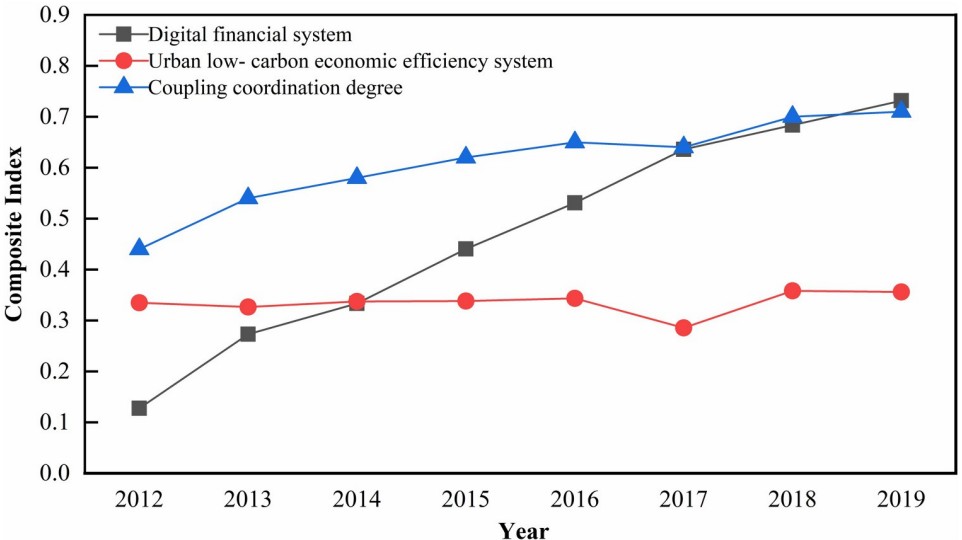

**Fig 3. Trend of the CCD.**

CCD showed a small decline in 2017, falling to 0.64. After 2017, CCD shows a continuous increase, growing to 0.71 in 2019.

**Spatial evolution of the CCD.** The evolution of coupling coordination levels from 2012 to 2019 is shown in Fig 4. The CCD covered a total of five levels in 2012: reluctant coordination, approaching imbalance, slight imbalance, severe imbalance, and extreme imbalance.

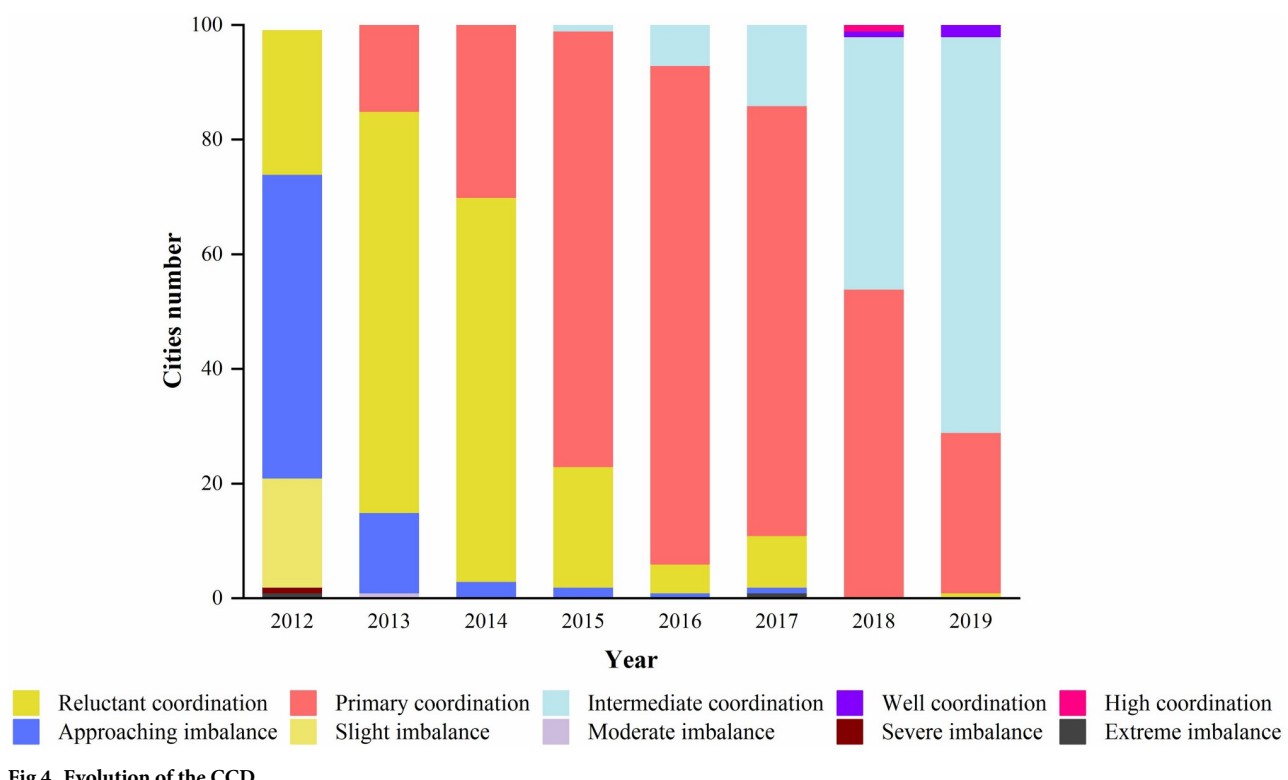

**Fig 4. Evolution of the CCD.**

Most of the cities in the YEB were in the state of approaching imbalance and slight imbalance, accounting for 72% of the number of cities studied, while the number of cities in the state of reluctant coordination only accounted for 25% of the total number, indicating that the coupling coordination between the two systems was poor in 2012 and needed further adjustment and improvement. There are five levels of coupling coordination in 2013, namely, primary coordination, reluctant coordination, approaching imbalance, and intermediate imbalance. Compared with 2012, the number of cities in reluctant coordination status increased significantly in 2013, from 25 cities in 2012 to 70 cities in 2013, and cities in primary coordination status began to appear among 100 cities, accounting for 16% of the total number of cities. The number of cities in approaching imbalance declined significantly, from 53 cities in 2012 to 14 cities in 2013, and the number of cities in slight imbalance dropped to 0. The coupling coordination spanned three levels in 2014: approaching imbalance, reluctant coordination, and primary coordination. The number of cities with approaching imbalance decreased further, accounting for only 3% of the total number. The number of cities in primary coordination status, increased by 14 cities compared to 2013. The cities in reluctant coordination state became the main part of 100 cities in YEB, accounting for 67% of the total number. In 2015, a small number of cities with intermediate coordination status began to appear, and cities with reluctant coordination status began to change to primary coordination status, which became the main component of the urban agglomeration in the YEB. After 2015, the cities with intermediate coordination increased year by year, while the cities with primary coordination decreased year by year, and the cities with well coordination status and high coordination status began to appear separately in 2018, and the cities with intermediate coordination status and primary coordination status became the main components of the urban cluster in 2019, accounting for 97% of the total number of cities.

In this paper, the years 2012, 2014, 2016, and 2019 were selected as time nodes. The spatial distribution characteristics of the CCD in the YEB are shown in Fig 5. The overall level of CCD of cities in the YEB was relatively low in 2012. Cities with the slight imbalance and approaching imbalance are primarily found in the upstream region of YEB, and cities with reluctant coordination are chiefly concentrated in the middle and downstream regions of the Yangtze River. Compared with 2012, the overall level of CCD of cities in the YEB improved in 2014, except for Chengdu, Zunyi, and Zhaotong, which are still in a state of approaching imbalance in the upstream region of the Yangtze River, most of the cities in approaching imbalance transformed into a state of reluctant coordination or primary coordination, mainly concentrated in the middle and lower reaches of the Yangtze River. In 2016, the coupling coordination of cities in the YEB was further improved, with some cities transforming from the slight coordination state to the intermediate coordination state, such as Suzhou, Zhenjiang, Quzhou, and Nanchang, which are mostly concentrated in the middle and lower Yangtze River region. The cities in reluctant coordination were mainly located in the upstream region of the Yangtze River, while the cities in slight coordination showed uniform distribution throughout the YEB, with a denser distribution in the middle and lower reaches of the Yangtze River. In 2019, most cities in the YEB were intermediately coordinated, and only a small number of slightly coordinated cities were located in the upper and middle reaches of the Yangtze River region. In summary, the CCD between ULEE and digital finance in China's YEB regions revealed a spatially growing distribution from upstream to downstream regions.

In order to further analyze the spatial shift of the center of gravity and the change of dispersion degree of urban coupling coordination in China's YEB, this paper used the SDE model to analyze the CCD in China's YEB in 2012, 2014, 2016, and 2019. The changes of relevant parameters of SDE are shown in Fig 6.

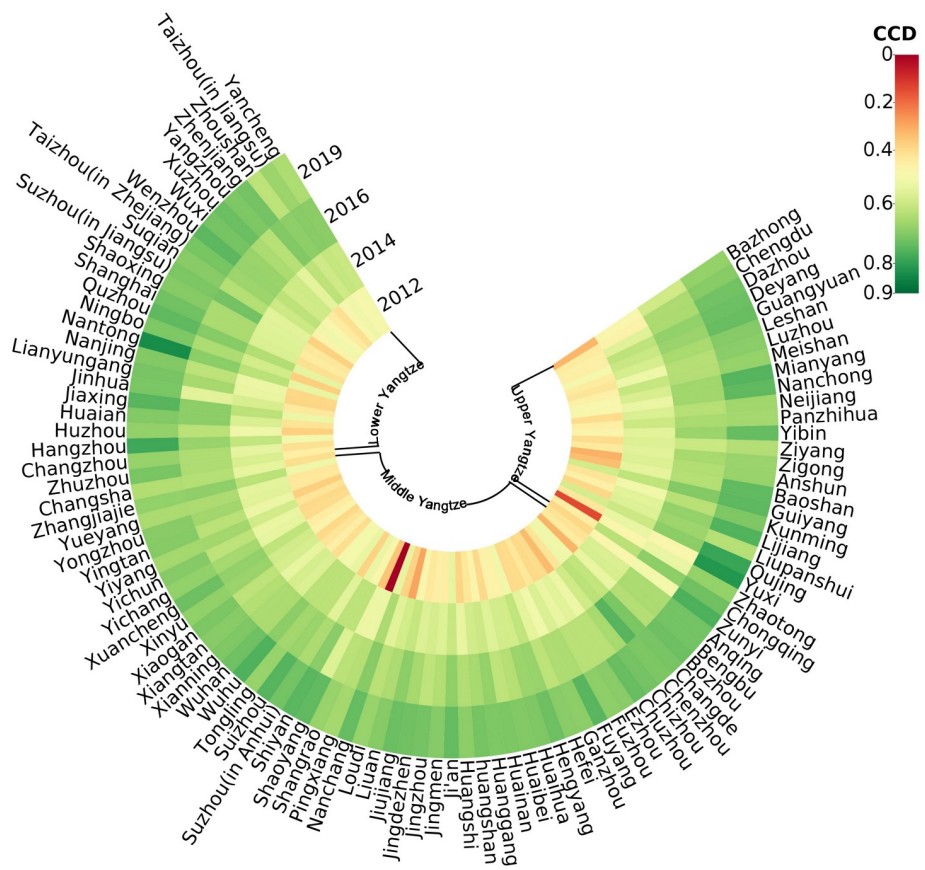

**Fig 5. Spatial distribution of the CCD.**

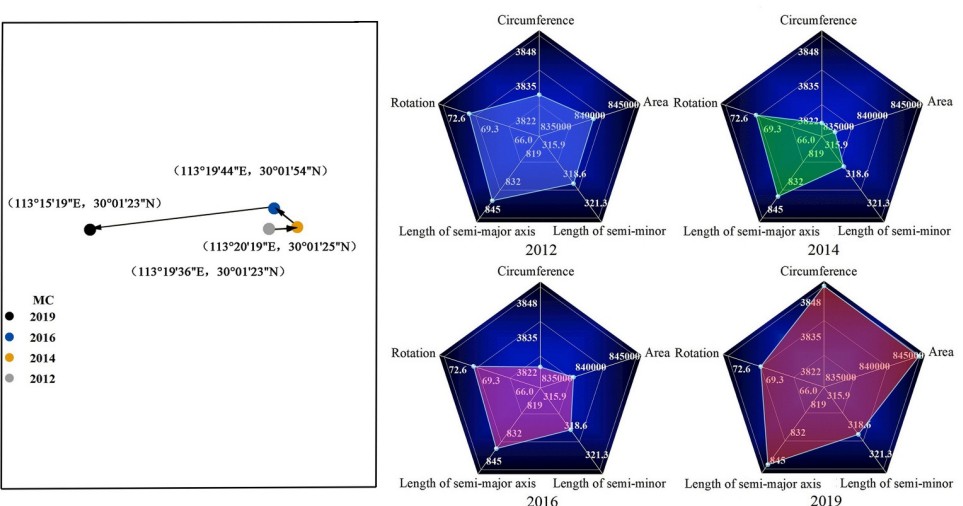

**Fig 6. The changes of relevant parameters of SDE.**

From the analysis results of the SDE model, we know that the center of gravity of urban CCD in China's YEB from 2012 to 2019 moved from (113°20'19 "E, 30°01'25 "N) to (113° 20'19 "E, 30°01'25 "N), and the range of movement was relatively small, which indicated that the spatial distribution pattern of urban CCD was stable during the study period. By period, the center of gravity of CCD shifted to the northeast direction in 2012–2014, the center of gravity of CCD shifted to the northwest direction in 2014–2016, and the center of gravity of CCD shifted to the southwest direction in 2016–2019.

In terms of the variation of the parameters of SDE, the length of the long semi-axis of SDE varied in the range of 838.03km~845.99km, the length of the short semi-axis varied in the range of 316.81km~318.40km, and the rotation angle of SDE varied in the range of 71.26°~ 71.91° from 2012 to 2019. The spatial distribution pattern of urban coupling coordination in the YEB was stable and dominated by the northeast-southwest direction. From 2012 to 2014, the distribution range of SDE showed an obvious narrowing trend, and the length of the long semi-axis and the rotation angle did not change significantly, while the length of the short semi-axis shortened slightly, indicating that the spatial distribution of the CCD of cities in the YEB tended to be concentrated, and the dispersion level of the CCD of cities distributed in the direction of the short axis of SDE decreased slightly. The rotation angle and the length of the long semi-axis remained basically unchanged from 2014 to 2016, while the length of the short semi-axis changed from a shortening trend in the previous period to an increasing trend, and the perimeter and area of the ellipse showed an increasing trend, indicating that the dispersion in the spatial distribution of the CCD increased and was mainly manifested in the direction of the short axis of the SDE. The SDE continued the previous trend of the area and perimeter expansion in 2016–2019, and the rate of expansion accelerated significantly. Both the long semi-axis length and the short semi-axis length of the ellipse showed an increase, with the increase in the long semi-axis length being particularly pronounced.

**Spatial pattern prediction of CCD.** It is difficult to study the problem of a small amount of data and poor information by general dynamic forecasting methods. In this paper, we introduced a gray model without excessive restrictions on the amount of modeling data to predict the coordinates of the center of gravity, the length of the long semi-axis, the length of the short semi-axis, and the rotation angle of the SDE respectively, to analyze the future changes of the spatial pattern of urban coupling coordination in the YEB in China. This paper uses the GM (1, 1) model in the gray model to construct a time series model for each parameter of the SDE, and the posterior difference test is used to test the confidence of the prediction results. The average relative error of the prediction results was 0.0362%, which was less than 1%. The average posterior difference ratio is 0.1875, which is less than 0.65. the average small error probability is 0.95, which is greater than 0.7. The development gray value is 0.0002, which is less than 0.3, and the correlation coefficient is 0.62, which is greater than 0.6. The above test results indicate that the model accuracy test is passed, and the prediction results are good and credible. The predicted results of the parameters of SDE are shown in Fig 7.

The prediction results expressed in Fig 7 show that the center of gravity of coupling coordination in 2019–2040 will move generally 22.17 km to the southwest, and the center of gravity will move 21.57 km in the east-west direction and 5.19 km in the north-south direction. The azimuth angle will rotate from 71.26° to 70.53°, indicating that the spatial distribution pattern of the coupling coordination will gradually shift from a predominantly northeast-southwest direction to a counterclockwise direction. From the perspective of spatial extent change, the long semi-axis of SDE will decrease from 845.99 km in 2019 to 842.03 km in 2040, a decrease of 0.47%. The short semi-axis of SDE will increase from 318.39 km in 2019 to 325.14 km in 2040, an increase of 2.12%. The area of SDE will grow from 846,068.98 km2 in 2019 to 860,923.16 km$^2$ in 2040.

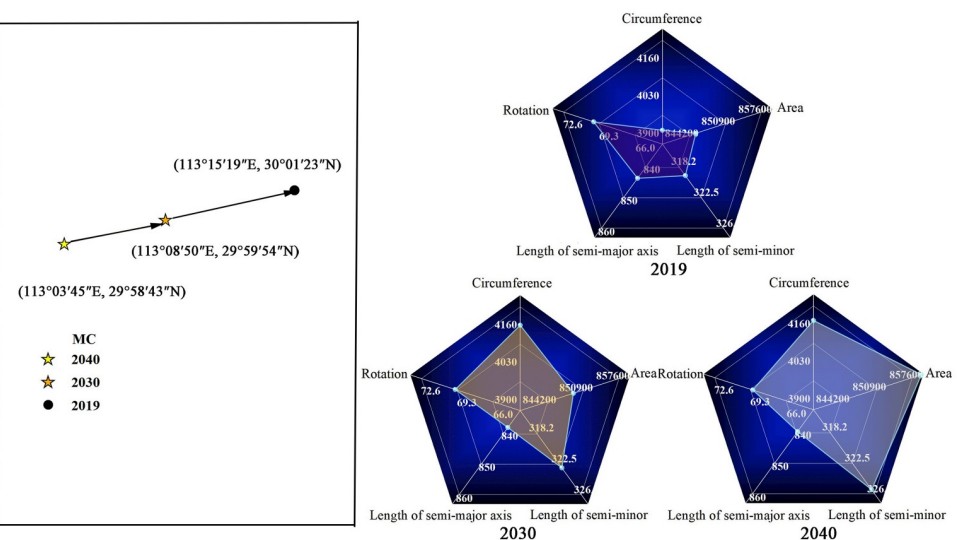

**Fig 7. Forecast of the spatial pattern.**

## Analysis of influencing factors

The CCD between ULEE and digital finance is influenced by various factors. In this paper, independent variables were selected from seven aspects: population development level, informatization level, greening level, transportation level, industrial structure, and scientific research level. The number of people per square kilometer (X1), the number of Internet broadband users (X2), the greening coverage of built-up areas (X3), the number of actual public-operated trams at the end of the year (X4), the share of tertiary industry in GRP (X5), and the number of employees in the research and comprehensive technical service industry (X6) were represented respectively. In this paper, the effects of missing data areas were excluded, and concerning the study of Wan (2020) [70], the six indicators were discretized separately, and the determinants q values and significance levels of each influencing factor on the coupled coordination of ULEE and digital finance were explored using the geographic detector. The results are shown in Table 5.

From a general perspective, population development level, informatization level, greening level, transportation level, industrial structure, and scientific research level are all drivers of

**Table 5. Results of coupling coordination influence factor detection.**

| Year | X1 | X2 | X3 | X4 | X5 | X6 |
|---|---|---|---|---|---|---|
| 2012 | 0.4701 | 0.3749 | 0.3698 | 0.3963 | 0.3447 | 0.5989 |
| 2013 | 0.3849 | 0.3668 | 0.4608 | 0.2982 | 0.5051 | 0.3718 |
| 2014 | 0.6608 | 0.6608 | 0.4593 | 0.5122 | 0.5102 | 0.5311 |
| 2015 | 0.4528 | 0.3323 | 0.5447 | 0.6105 | 0.5277 | 0.4495 |
| 2016 | 0.5367 | 0.2801 | 0.5274 | 0.5779 | 0.6335 | 0.4744 |
| 2017 | 0.3843 | 0.4120 | 0.3682 | 0.8472 | 0.4315 | 0.3995 |
| 2018 | 0.4687 | 0.4593 | 0.3997 | 0.3267 | 0.2259 | 0.5418 |
| 2019 | 0.4419 | 0.4684 | 0.3431 | 0.3729 | 0.4034 | 0.4627 |

Note: all the above results passed the 1% significance test.

CCD in ULEE and digital finance in the YEB region of China. Among them, the influence of informatization level and industrial structure on CCD tends to increase gradually, while the influence of population development level, greening level, transportation level, and scientific research level on CCD gradually decreases. Specifically, the impact of the level of population development was slowly declining overall, with an average annual decrease of 0.35%. However, the influence of this factor on CCD was still at a relatively high level compared to other factors. The influence of this factor on CCD gradually increased in 2012–2014, reaching the maximum in 2014, and then decreasing in 2015–2019. The impact of the informatization showed an overall trend of growth, with an average annual growth of 1.17%. The factor had the most significant impact on the coupling coordination in 2014, and the smallest impact was demonstrated in 2016. The overall impact of greening level showed a slow downward trend, with an average annual decrease of 0.33%. The impact showed an increasing trend from 2012 to 2015, and then the impact gradually declined. The overall impact of the level of transportation showed a slowly decreasing trend, with an average annual decrease of 0.29%, with a gradual increase in impact in 2012–2017, followed by a trend of significantly lower impact. The impact of the industrial structure showed an overall upward trend with an average annual increase of 0.73%, with the factor q reaching its highest value in 2016 and its lowest value in 2018. The influence of the scientific research level showed a decreasing trend, with an average annual decrease of 1.70% and q-values fluctuating between 0.3715–0.5989, and the q-value of this factor was still relatively high compared to other factors.

## Discussion

This paper measured the ULEE of 100 cities in the YEB region of China and then analyzed the spatial and temporal evolutionary trends of the coupled coordination of ULEE and digital finance and the changes in the main influencing factors in conjunction with the Peking University Digital Inclusive Finance Index. These findings can help policy research departments to understand the coordinated development relationship between ULEE and digital finance in the YEB region of China in recent years, to better promote the two systems to achieve quality and harmonious development.

First, from a time evolution perspective, ULEE showed an overall fluctuating growth trend from 2012–2019, with a total increase of 7.41%. On the one hand, PEC and SEC grown by 13.16% and 4.8% respectively, indicating that the growth of ULEE is mainly driven by PEC and SEC. This is because the Chinese government has formulated a set of policies to boost the growth of the low-carbon economy since 2012 to encourage enterprises to use clean energy and improve energy use efficiency, which has led to the widespread and in-depth use of carbon reduction technologies in cities in the YEB region [71]. On the other hand, the TC declined, which also indicates that the lack of science and technology innovation in the YEB region has limited the further progress of the efficiency of low-carbon economic development in the region. However, the TC has shown a significant increase in 2015, which may be attributed to the Chinese government's enhanced financial support and further implementation of tax breaks for low-carbon industries in 2015, which has boosted low-carbon technological innovation in enterprises [72]. Therefore, promoting low-carbon production technology innovation in the R&D sector and developing new clean energy sources will be an important direction for the growth of the low-carbon economy in cities in the YEB region in the future. In addition, ULEE started to show continuous and stable growth after 2017, which may be related to the official launch of the carbon emission trading market at the end of 2017. Under the market mechanism, market players with excess or insufficient allowances can meet their emission reduction targets more effectively through trading. In other words, the carbon trading

mechanism reduces the cost of carbon reduction for enterprises and accelerates the innovation and diffusion of low-carbon technologies [73], thus promoting the optimization of industrial structure and realizing the efficiency of the low-carbon economy. From the perspective of spatial distribution, the upstream region had the highest ULEE, which mainly benefited from the efficient application of low-carbon technologies and the spillover of scale dividends, but there is still room for improvement in technological innovation in this region. The midstream region had the lowest PEC among the three regions, and its low-carbon economy was at the scale development stage. Therefore, the midstream region should accelerate low-carbon technology innovation and strengthen the application and diffusion of low-carbon technologies. The downstream region outperformed the other regions in terms of low carbon technology innovation, but had the lowest level of SEC, indicating that the downstream region had the relatively least scale dividend. From the perspective of city characteristics, Yuxi, Jingzhou, Zunyi, Lijiang, Qujing, Guangyuan, Changzhou, and Shanghai had high levels of ULEE among the YEB regions in China. Among them, Yuxi, Jingzhou, Zunyi, Lijiang, Qujing, and Guangyuan, inspired by China's high-quality development goals, have rapidly developed low-carbon tourism and significantly increased their fiscal revenues, thus implementing a set of policies to boost energy restructuring, including rational use of natural superiorities to exploit low-carbon energy, improve energy use efficiency and vigorously promote low-carbon lifestyles to the public, which have greatly enhanced the cities' ULEE. As for Changzhou and Shanghai, both cities have relatively sound infrastructure, reasonable industrial structures, convenient transportation, a concentration of talents, and a strong capacity for low-carbon technological innovation. Shanghai, in particular, as the world's financial center, has a high level of digital financial development, which offers efficient financial support for the development of green technologies and is contributive to the study, development, and promotion of new clean energy sources, which can significantly improve the city's ability to reduce carbon emissions. The ULEE of Zhaotong, Taizhou, Ziyang, Huzhou, Jiaxing, Dazhou, Changsha, Fuzhou, Nanchang, Yichang, Wenzhou, Liu'an, Loudi, and Jingmen was at a relatively low level in the YEB zone, primarily due to the low level of TC in these cities, and therefore the innovation and application of low-carbon technologies should be strengthened. The low PEC in Xianning and Yichun restricted the improvement of ULEE, so optimizing the application of technology and improving the management of technology should be the focus of future development in these cities. The mean values of SEC in Ji'an and Jingmen were less than 1. They need to expand their production scale and increase factor inputs for low-carbon economic development, to improve ULEE. In summary, this paper here discussed the spatio-temporal evolution characteristics of ULEE in China's YEB region from 2012–2019, answering the first question raised in "Introduction" section.

Second, from the perspective of time evolution, CCD showed an overall trend of stable growth from 2012–2019. Specifically, the CCD grew slowly from 2012–2016. This is mainly because digital finance was still in its infancy and the intensity of support for the growth of the low-carbon economy was insufficient, resulting in slow growth of ULEE and thus slowing down the coordinated development of ULEE and digital finance [74]. CCD has continued to grow after 2017, probably due to the rapid growth of China's digital finance sector and the launch of China's carbon emission rights market, which have stimulated the advance of low-carbon industries and the restructuring of energy consumption, promoting urban carbon reduction [75], thus allowing for a higher level of coordination between the two systems. From the perspective of spatial evolution, most of the cities in the YEB experienced a shift from approaching imbalance and slight imbalance to primary coordination and intermediate coordination during the study period, which indicates that the coordinated development relationship between ULEE and digital finance in the YEB region has been widely improved. Among

them, the lower Yangtze River region had the highest degree of digital finance development relative to other regions, allowing cities to receive more abundant and efficient financial support in the process of low-carbon economic development, thus promoting clean technology research and development and improving the energy consumption structure to achieve a high level of CCD development. The cities in the upstream region are mostly inland cities, where access to resources such as labor, technology, and capital is difficult, and production and operation are mostly based on crude industrial production, making it difficult for the urban economy and urban environment to develop in a coordinated manner [76]. Meanwhile, the growth of digital finance in the region is comparatively lagging, with insufficient financing support, making it hard for the two to accomplish coordinated evolution, and therefore the CCD level in the region was the lowest. The value of the CCD in the middle reaches of the Yangtze River lies between the CCD in the upper reaches and those in the lower reaches. In addition, the cities in the YEB region with a relatively high level of coordinated development of ULEE and digital finance are mainly located in the northeast-southwest direction. The spatial center of gravity shifted first to the northeast, then to the northwest, and finally to the southwest during the study period. This is mainly because the CCDs of cities such as Liu'an, Zhenjiang, and Yancheng, which are located in the northeast direction, showed a significant increase from 2012–2014. CCDs in cities such as Mianyang and Chengdu, located in the northwest direction, increased significantly from 2014–2016. CCDs in cities in the southwest of the YEB increased at a faster rate after 2016, while CCDs in cities in the northeast maintained a relatively low rate of growth. Thus, a trajectory of the center of gravity moving first to the northeast, then to the northwest, and finally to the southwest was formed. In terms of the future direction of the spatial pattern, the spatial center of gravity of CCD will move southwestward in the middle region of the YEB in 2019–2040, indicating that the western cities in the middle Yangtze River region are expected to boost the coordinated growth of ULEE and digital finance in the future. From 2019 to 2040, the discrete level of CCD in the northeast-southwest direction will show a downward trend, while the discrete level in the northwest-southeast direction will increase, indicating that cities in the northwest-southeast direction have the possibility of rapidly increasing CCD, and the exchange of low-carbon technologies, financial resources, digital technologies and technical talents between regions will be further strengthened, thus benefiting cities with relatively low CCD levels. In addition, the spatial pattern of CCD in 2019–2040 is dominated by the northeast-southwest direction and will turn in a counterclockwise direction, again indicating that CCD in cities with a northwest-southeast direction has a strong potential to improve in the future. In summary, this paper here discusses the spatio-temporal evolutionary characteristics of the CCD of ULEE and digital finance from 2012–2019 and the future spatial pattern changes, answering the second question posed in "Introduction" section.

Third, in terms of influencing factors, the level of informatization and industrial structure tended to gradually increase their influence on CCD, while the level of population development, greening, transportation, and scientific research gradually weaken their influence on CCD. In terms of the influence of the level of population development, the influence of population development showed a slow downward trend overall, with an average annual decrease of 0.35%. However, the influence of this factor was still at a relatively high level compared to other factors. This is mainly because as the number of people increases, there is a corresponding increase in social labor resources and consumption behavior of the population, which helps to boost the efficiency of economic development while promoting the development of the digital financial sector. Changes in low-carbon consumption behavior would also have a vital influence on carbon emissions. As a result, the influence of the level of population development on CCD was slowly diminishing but still at a high level. In terms of the impact of the level of informatization, the impact of the level of informatization showed an overall trend of

growth over time, with an average annual growth of 1.17%. This is because the popularity of the Internet is the basis for the creation and advance of digital finance, and digital financial services need to be realized through the Internet and related information technology. And the construction of information technology has given impetus to the birth of more new green industries, providing new impetus to the growth of the low-carbon economy. In terms of the impact of greening levels, the overall impact of greening levels showed a slow downward trend, with an average annual decrease of 0.33%. This may be because with the deepening of greening construction, the space available for greening construction in built-up areas gradually becomes less and the marginal contribution of greening level to the low-carbon economy gradually decreases, and the driving effect of greening construction on the low-carbon economy diminishes, thus obstructing the coordinated advance of ULEE and digital finance. In terms of the influence of transport levels, the influence of this influence showed an overall decreasing trend, with the influence increasing year on year from 2012–2017, but decreasing sharply after 2017, which should be attributed to the increase in the number of public-operated trams from 2012–2017, the increase in the number of low-carbon trips made by urban residents and the corresponding increase in urban transport capacity, which led to a good development of urban low-carbon economic efficiency, but in recent years the number of public-operated trams in the city gradually saturated and the number of private cars in the city increased sharply, boosting a raise in $CO_2$ emissions. In terms of the influence of industrial structure, the influence of this factor on CCD increased gradually, indicating that the Chinese government issued relevant policies to promote the proportion of tertiary industries in GDP, eliminating industries related to high pollution, high energy consumption, and high carbon emissions, and services such as digital finance have been better developed, and further improvement of industry construction is an important direction to coordinate the development of ULEE and digital finance. In terms of the influence of the level of scientific research, the influence of the level of scientific research on CCD decreased overall, but its influence was still at a high level compared to other factors, which indicates that improving the level of scientific research and innovation is still a priority for the development of cities in the YEB region of China and that cities should actively accelerate the upgrading of energy conservation and digital information technology, while ensuring that cities make effective use and promotion of existing technology, fully applying advanced technology to the growth of low carbon economies in cities, and accelerating the construction of digital financial services so that digital finance can provide strong financial support for the growth of the low carbon economy in the future. In summary, this paper discussed here the impact of factors such as level of population development, level of informatization, level of greening, level of transportation, industrial structure, and level of scientific research on the CCD of ULEE and digital finance in the YEB region of China from 2012–2019, answering the third question posed in "Introduction" section.

Besides the above answers to the research questions, this paper provided several novel contributions to related studies. Firstly, in terms of ULEE evaluation, this paper complemented the framework for evaluating low-carbon economic efficiency in previous studies [18, 50]. Compared with existing researches, this paper improved the evaluation framework of low carbon economy efficiency in terms of land use and financial support. Other studies on low-carbon economy efficiency can be further expanded based on this, such as the interaction between low-carbon economy efficiency and the level of technological development [77]. Secondly, this study provided a new quantitative analysis perspective and is the first study to analyze the changes in the level of coordinated development of ULEE and digital finance through a coupled coordination degree model, which helps to help relevant sectors understand the coordinated development relationship between ULEE and digital finance. Thirdly, this paper further analyzed the expected changes in the spatial pattern of CCD between ULEE and digital

finance and the influencing factors, which will provide a basis for local governments in China to formulate policies to enhance the level of coordinated development of ULEE and digital finance.

In addition, there are still some limitations to this paper. Firstly, the paper uses the latest data that can be collected at present, but there is still a certain lag, for further research can be done after the data is updated. Secondly, this paper does not take into account the climate change factor in the ULEE evaluation framework, and the model can be expanded in future research to improve the ULEE evaluation framework. Thirdly, this paper only considers the effect of individual factors on the level of coordinated growth of ULEE and digital finance but does not analyze the interaction of multiple influencing factors, which can be discussed in more depth in future research.

## Conclusion and countermeasures

### Conclusion

The coupled and coordinated relationship between ULEE and digital finance is of critical significance for urban sustainability. In this paper, the comprehensive assessment index systems of the two major urban systems were established respectively, taking 100 cities in the YEB of China as examples. Through CCD model, SDE model, GM (1, 1) model, and geographic detector, the spatial and temporal change of coupled and coordinated development of the two systems, future spatial pattern changes, and influencing factors were studied to provide an effective theoretical basis for strengthening the coupled and coordinated relationship between low-carbon economy efficiency and digital finance of cities in China's YEB. The specific conclusions are as follows:

1. The ULEE of China's YEB region showed a slow fluctuating upward trend from 2012 to 2019, and the ULEE showed a spatial distribution of the strongest in the upstream region, the second strongest in the downstream region, and the weakest in the midstream region. From a temporal perspective, the ULEE in China's YEB maintains an average annual growth of 0.93% during the study period. After the decomposition of indicators, it can be seen that PEC and SEC grew by 13.16% and 4.80%, respectively, during the study period, which drove the growth of ULEE. From a spatial perspective, 74 out of 100 cities had ULEE values above 1, among which Yuxi, Jingzhou, Zunyi, Shanghai, Lijiang, Changzhou, Qujing, and Guangyuan had relatively high ULEE. The other 26 cities still had more potential for boosting in terms of economic growth and carbon emission control.

2. The CCD of ULEE and digital finance in China's YEB region maintained a stable growth trend from 2012 to 2019, and the CCD showed a spatially increasing distribution from upstream to downstream. From a temporal perspective, urban CCD maintained an average annual growth of 3.42% from 2012 to 2019, from 0.4371 in 2012 to 0.7108 in 2019. From a spatial perspective, the main hierarchical types of urban CCD in the YEB region changed from approaching imbalance and slight imbalance to primary coordination and intermediate coordination from 2012 to 2019. The CCD in the YEB region shows an overall spatial distribution of gradient from the upstream region to the downstream region from 2012 to 2019. The spatial center of gravity shifts from (113°20'19 "E, 30°01'25 "N) to (113°20'19 "E, 30°01'25 "N), and the spatial pattern is dominated by the northeast-southwest direction, with the long semi-axis of SDE growing from 838.03km to 845.99km and the short semi-axis growing from 316.81km to 318.40km. The overall level of dispersion of the CCD shows an overall trend of expansion.

3. The future spatial pattern of the CCD of ULEE and digital finance in the YEB region of China will be dominated by the northeast-southwest direction and shift in the counter-clockwise direction, and the dispersion level will show different trends in different directions. It is predicted that during 2019–2040, the spatial center of gravity of the CCD of the two major urban systems in the YEB region of China will move 22.17 km to the southwest. The rotation angle of SDE will change from 71.26˚ in 2019 to 70.53˚ in 2040, the long semi-axis of SDE will decrease by 0.47% and the short semi-axis will increase by 2.12%. The spatial distribution pattern of the CCD will be dominated by the northeast-southwest direction and will shift in the counterclockwise direction, with the overall trend of dispersion in the northwest-southeast direction and the overall trend of insignificant concentration in the northeast-southwest direction.

4. The coupling and coordination of low-carbon economy efficiency and digital finance in cities of the YEB in China were influenced by various factors and their influence varied during the study period. On the whole, the factors affecting the CCD of the two systems in cities were: population development level, informatization level, greening level, transportation level, industrial structure, and research level. Among them, the influence of informatization level and industrial structure showed an overall trend of increasing during the study perio, with q-values increasing by 1.17% and 0.73%, respectively. The influence of population development level, greening level, transportation level, and scientific research level showed a gradual weakening overall, with q-values decreasing by 0.35%, 0.33%, 0.29%, and 1.70%, respectively, but the influence of population development level and scientific research level was still at a high level compared with other drivers.

## Countermeasures and suggestions

According to the above findings, it can be learned that the CCD between ULEE and digital finance in China's YEB is generally at a good level, but there are still some problems. To further enhance the coordinated development of the two systems in cities, this paper proposed the following three policy recommendations:

1. Clarify the path to improve the ULEE. At present, the low-carbon economy efficiency of a few cities in the YEB has still not reached the effective frontier. Cities with weaker technological progress should increase investment in scientific and technological research and development, accelerate research and development of new low-carbon technologies, reasonably develop new clean energy sources, and effectively improve the efficiency of energy utilization. Cities with weak technical efficiency should optimize factor utilization on time, actively adjust the ratio of resource factor inputs and outputs, and improve the use of energy saving and emission mitigation technologies to realize sustainable development. For cities with insufficient scale efficiency, the local government should implement relevant policies, give financial support and policy preferences to relevant green industries, encourage the increase of factor inputs and expand production scale to achieve the rapid development of low-carbon industries, to pull the ULEE. In addition, regions with high low-carbon economy efficiency should fully exploit the radiation effect, strengthen the rational allocation of resource factors, and promote the synergistic development of ULEE among regions.

2. Strengthen the interconnection between digital finance and ULEE and realize the coordinated development of the two systems. Financial institutions should actively strengthen the digital development process, effectively broaden low-carbon economy data and manage them effectively, increase support for low-carbon and green industries in the index design

of credit business, improve the accuracy of digital financial services, and effectively contribute to the development of ULEE. Meanwhile, financial institutions should improve their products with the help of Internet of Things technology, and reasonably monitor and track the carbon emissions of key low-carbon enterprises to reduce credit risks. The credible and large amount of carbon emission information can be a guide to the ULEE. In addition, the rapid development of the low carbon economy will accelerate the progress of information technology and the traditional finance industry, thus promoting the development of digital finance.

3. Enhance the positive influence of influencing factors on the CCD of the two systems in cities. The government should strengthen the management of the population to ensure that the city has an abundant labor force for economic development, while continuously improving the education level of the population and popularizing the low-carbon concept and digital technology application. The government should also focus on the transportation and greening level of the city, support the use of low-carbon energy in transportation, and reasonably increase the greening area of the city to enhance the development momentum of the city's low-carbon economy. In addition, the government should also intensify investment in scientific research, boost the advancement of low-carbon technologies and information technology, actively promote the upgrading of industrial structure, and hasten the development of ULEE and digital finance to achieve sustainable urban development.

## Supporting information

**S1 Table. Urban low-carbon economy efficiency and decomposition index from 2012 to 2019.**
(DOCX)

**S2 Table. Coupling coordination degree of urban low carbon economy efficiency and digital finance from 2012 to 2019.**
(DOCX)

## Author Contributions

**Data curation:** Fengge Yao.

**Software:** Liqing Xue.

**Writing – original draft:** Jiayuan Liang.

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
