## [Decision Letter · Decision Letter 0]

21 Mar 2022

PONE-D-22-00910Research on Coupling Coordination and Influencing Factors between Urban Low-carbon Economy Efficiency and Digital Finance – Evidence from 100 Cities in China’s Yangtze River Economic BeltPLOS ONE

Dear Dr. Liang,

Thank you for submitting your manuscript to PLOS ONE. After careful consideration, we feel that it has merit but does not fully meet PLOS ONE’s publication criteria as it currently stands. Therefore, we invite you to submit a revised version of the manuscript that addresses the points raised during the review process.

We look forward to receiving your revised manuscript.

Kind regards,

Xingwei Li, Ph.D.

Academic Editor

PLOS ONE

Journal Requirements:

“1、（1）Initials of the authors who received each award：Yao Fengge

（2）Grant numbers awarded to each author：17BJY119

（3）The full name of each funder：The National Social Science Foundation of China Project

（4）URL of each funder

website：http://fz.people.com.cn/skygb/sk/index.php/Index/seach

  2、（1）Initials of the authors who received each award：Liang Jiayuan

（2）Grant numbers awarded to each author：YJSCX2020-624HSD

（3）The full name of each funder： The Harbin University of Commerce 2020

Postgraduate Innovation Research Funding Project

（4）URL of each funder website：https://yjsc.hrbcu.edu.cn/

3、Did the sponsors or funders play any role in the study design, data collection and analysis, decision to publish, or preparation of the manuscript? Yes.The funders provided the cost of data collection and the software needed for the study.”

We note that you have provided additional information within the Funding Section that is not currently declared in your Funding Statement. Please note that funding information should not appear in the Funding section or other areas of your manuscript. We will only publish funding information present in the Funding Statement section of the online submission form.

“This work was supported by the National Social Science Foundation of China (17BJY119) and the Harbin University of Commerce 2020 Postgraduate Innovation Research Funding Program (YJSCX2020-624HSD).”

4. We note that Figures 1, 3, 6, 7 and 9 in your submission contain [map/satellite] images which may be copyrighted. All PLOS content is published under the Creative Commons Attribution License (CC BY 4.0), which means that the manuscript, images, and Supporting Information files will be freely available online, and any third party is permitted to access, download, copy, distribute, and use these materials in any way, even commercially, with proper attribution. For these reasons, we cannot publish previously copyrighted maps or satellite images created using proprietary data, such as Google software (Google Maps, Street View, and Earth). For more information, see our copyright guidelines: http://journals.plos.org/plosone/s/licenses-and-copyright.

   1. You may seek permission from the original copyright holder of Figures 1, 3, 6, 7 and 9 to publish the content specifically under the CC BY 4.0 license.  

Maps at the CIA (public domain): https://www.cia.gov/library/publications/the-world- factbook/index.html and https://www.cia.gov/library/publications/cia-maps-publications/index.html

Additional Editor Comments (if provided):

All reviewers have pointed out significant flaws in this manuscript. Please carefully revise your manuscript in accordance with these suggestions.

Reviewers' comments:

Reviewer's Responses to Questions

**Comments to the Author**

1. Is the manuscript technically sound, and do the data support the conclusions?

Reviewer #1: Partly

Reviewer #2: Yes

2. Has the statistical analysis been performed appropriately and rigorously? 

Reviewer #1: Yes

Reviewer #2: Yes

3. Have the authors made all data underlying the findings in their manuscript fully available?

Reviewer #1: No

Reviewer #2: Yes

4. Is the manuscript presented in an intelligible fashion and written in standard English?

Reviewer #1: Yes

Reviewer #2: Yes

5. Review Comments to the Author

Reviewer #1: 1. While the author presents the Abstract, answer the questions carefully: What problem did you study and why is it important? What methods did you use? What were your main results? And what conclusions can you draw from your results? Please make your abstract with more specific and quantitative results while it suits broader audiences. Frankly, the content has too many abbreviations, which makes the manuscript hard to follow. It is suggested that author can refine this part.

2. The current Introduction should be further improved. A good one includes at least four aspects: motivation/background, literature review, aim and contribution, and organization of the remains of the study. Avoiding to put massive bibliographies behind one sentence. Such as XXXXX [1-5], OR 1, 2, 3, 4, 5; all references should be cited with detailed and specific descriptions.

3. Literature Review has the chance to be further improved: it seems that the authors have made the retrospection. However, via the review, what issues should be addressed? What is the current specific knowledge gap? What implication can be referred? The above questions should be point-by-point answered clearly. This work focused on the low carbon transition in economic growth in 100 cities. At the consumption side, buildings show the most significant potential in cost-effectiove emission reduction, which is worthy to be discussed. Here, some latest literature investigating the low carbon roadmap or carbon neutral pathway of building sector is worhy to added and discussed. E.g., DOI: 10.1016/j.eneco.2021.105712 DOI: 10.1016/j.apenergy.2021.118098 DOI: 10.3390/buildings12010054 DOI: 10.3390/buildings12010083 DOI: 10.3390/buildings12020128

4. For some details, be sparing in the use of tables and ensure that the data presented in them do not duplicate results described elsewhere in the article. It is suggest to avoid using vertical rules and shading in table cells. It is also suggested that the key findings should be summarized in Conclusion one by one with the marks of 1. 2. 3. or i. ii. iii. etc.

5. As a key part of a paper, Discussion should show the readers at least two elements: "breadth" and "depth". "Breadth" reflects whether the analytical results can be explained via different approaches. "Depth" reflects whether the analytical results completely answer the questions raised in Introduction. My first sense shows the current Discussion is without enough insight. This should explore the significance of the results of the work, not repeat them. A combined Results and Discussion section is OK. However, avoid extensive citations and discussion of published literature.

Reviewer #2: The aim of this manuscript is to research coupling coordination and influencing factors between ULEE and digital finance. It is an interesting topic. There are some comments of improvement in the manuscript:

1. Recommended to add some quantitative analysis to the abstract.

2. The contribution and innovation of manuscript can appropriately add some dialogues with the past literature.

3. Figures need to improve clarity.

4. Suggested to supplement the energy conversion factor value used in the calculation of carbon emissions.

5. Discussion is needed. By interacting with previous studies, the author should discuss more about the "story" behind the results and data.

6. Originality and novelty of the paper needs to be further improved and clarified.

7. Writing needs to be significantly improved.

6. PLOS authors have the option to publish the peer review history of their article (what does this mean?). If published, this will include your full peer review and any attached files.

Reviewer #1: No

Reviewer #2: No

---

## [Author Response · Author response to Decision Letter 0]

3 May 2022

Dear Editors and Reviewers,

Thank you very much for your kindly comments concerning our manuscript entitled “Research on Coupling Coordination and Influencing Factors between Urban Low-carbon Economy Efficiency and Digital Finance -- Evidence from 100 Cities in China's Yangtze River Economic Belt.” (PONE-D-22-00910). Your comments and those of the reviewers are all valuable and very helpful for revising and improving our paper, as well as the important guiding significance to our researches. We have studied comments carefully and have made correction which we hope meet with approval. Revisions in the text are shown using yellow highlight [example] for additions, and strikethrough font [example] for deletions. We hope that the revisions in the manuscript and our accompanying responses will be sufficient to make our manuscript suitable for publication in PLOS ONE. The main corrections in the paper and the responds to the reviewer’s comments are as flowing.

Responds to the Journal’s comments:

1.Please ensure that your manuscript meets PLOS ONE’s style requirements, including those for file naming. 

Response: We fully respect the requirements of the journal and we have modified the format of the article according to the requirements of the journal, hoping to meet the PLoS One’s style requirements.

2.The question in the Funding section.

Response: We fully respect the requirements of the journal and the funding information should not appear in the Funding section or other areas of the manuscript. Therefore, we will apply for the journal to remove any funding related words from the manuscript on our behalf in the cover letter by deeply greatful. We have removed any funding related words in the revised manuscript. Our revised statement is as follows:

This work was supported by the National Social Science Foundation of China (17BJY119) and the Harbin University of Commerce 2020 Postgraduate Innovation Research Funding Program (YJSCX2020-624HSD).The funders provided the cost of data collection and the software needed for the study.

3.In your Data Availability statement, you have not specified where the minimal data set underlying the results described in your manuscript can be found.

Response: We fully respect the requirements of the journal and We have uploaded the minimum base dataset of the study as a supporting information file. we add where to find the minimum data set of the results described in the Data Availability statement, the contents are as follows:

The minimum data set for our manuscript is derived from the China Urban Statistical Yearbook, which is available to the general public. The link to the database is https://data.cnki.net/yearbook/Single/N2022040095

4.We note that Figures 1, 3, 6, 7 and 9 in your submissions contain map images which may be copyrughted. 

Response: We fully respect the requirements of the journal and as for the copyright protection of the maps in the manuscript, we have completely deleted the map in Figure 1 and 7 in the manuscript, and changed the map shown in Figure 3 in the empirical analysis to bubble plot, which may highlight the differences of the urban low-carbon efficiency between regions. We also changed the map shown in Figure 6 to heat map, which can reflect the spatial variability of the coupling coordination. We replaced the map in Figure 9 with a radar map that adequately represents the future spatial pattern changes of the coupled coordination. In addition, we made adjustments to Figure 8 of the original manuscript and added the trajectory of the spatial center of gravity .

Responds to the additional editor’s comments:

1.All reviewers have pointed out significant flaws in this manuscript. Please carefully revise your manuscript in accordance with these suggestions.

Reponse: We humbly accepted the comments of the additional editor, fully considered the comments of reviewer #1 and reviewer #2. In accordance with the reviewers’ suggestion, we have made the following main changes to the article:

1.we added quantitative, concrete results of the analysis in the “Abstract” section.

2.We added three questions that are the focus of this paper in the “Introduction” section, and further clarified the main contributions of this paper.

3.We summarized specific current knowledge gaps in the “Literature review” section, as well as described the specific work done in this paper to close these gaps.

4.We supplemented the "Literature Review" section with five papers on low carbon emission reduction in the construction industry recommended by reviewer 1.In the “Results” section, we provided a more detailed analysis, discussed the reasons for the results, and answered each of the questions posed in the “Introduction” section.

5. In the “Conclusion and countermeasures” section, we provided a more comprehensive summary of the paper's findings.

6.In terms of details. We added the values of energy conversion coefficients used in calculating carbon emissions, improved the clarity of the images in the article, removed and modified images with maps from the original draft ,removed Table 5 , and presented the contents in textual form instead, to save the table. 

Responds to the reviewer’s comments:

Responses to the comments of reviewer #1

1. While the author presents the Abstract, answer the questions carefully: What problem did you study and why is it important? What methods did you use? What were your main results? And what conclusions can you draw from your results? Please make your abstract with more specific and quantitative results while it suits broader audiences. Frankly, the content has too many abbreviations, which makes the manuscript hard to follow. It is suggested that author can refine this part.

Response: Thank you for your comments and suggestion concerning our manuscript. The comments and suggestions are all valuable and very helpful for revising and improve our paper, as well as the important guilding significance to our researches. We have studied comments carefully and have made correction which we hope meet with approval. 

The following are our answers to the questions you asked: 

1.Our study problem is the coupled and coordinated relationship between urban low-carbon economy efficiency and digital finance and its influencing factors. 

2.China is a large carbon emitting country, and improving urban low-carbon economy efficiency is conducive to reducing CO2 emissions while maintaining stable urban economic growth. Digital finance is an important financing channel for urban low-carbon economic transformation and an effective way for the financial industry to carry out low-carbon development. The connection between urban low-carbon economic efficiency and digital finance is getting stronger and stronger, and the interaction between the two systems is becoming more and more obvious. Analyzing how to promote the coordinated development of urban low-carbon economic efficiency and digital finance is of vital importance for sustainable urban development, so it is important to analyze the coupled and coordinated relationship between urban low-carbon economy efficiency and digital finance and its influencing factors. 

3.We use the global Marquist-Luneberg index model, the coupling coordination degree model, the standard deviation ellipse model, the gray model and the geodetector model as methods. Among them, GML index model is used to measure urban low carbon economic efficiency, coupling coordination degree model is used to measure the coupling coordination degree of urban low carbon economy efficiency and digital finance, standard deviation ellipse model is used to analyze the spatial pattern change of coupling coordination degree, gray model is used to predict the future spatial distribution change of coupling coordination degree, and geographic detector model is used to analyze the influencing factors of coupling coordination degree.

4.Our results are mainly the following four points: (1) The urban low-carbon economy efficiency of China's Yangtze River economic belt shows a slow fluctuating upward trend from 2012 to 2019, and the urban low-carbon economy efficiency shows a spatial distribution of the strongest in the upstream region, the second strongest in the downstream region, and the weakest in the midstream region. From a temporal perspective, the low-carbon economy efficiency of cities in China's Yangtze River Economic Belt maintains an average annual growth of 0.93% during the study period. After the index decomposition, it can be seen that the pure technical efficiency and scale efficiency have increased by 13.16% and 4.80%, respectively, during the study period, which drives the growth of urban low-carbon economy efficiency. From a spatial perspective, urban low-carbon economy efficiency has variability in spatial distribution, with 74 out of 100 cities having urban low-carbon economy efficiency values above 1. Among them, Yuxi, Jingzhou, Zunyi, Shanghai, Lijiang, Changzhou, Qujing, and Guangyuan have relatively high urban low-carbon economy efficiency. The other 26 cities still have more room for improvement in terms of economic development and carbon emission reduction. (2) The coupling coordination between urban low-carbon economy efficiency and digital finance in China's Yangtze River Economic Belt maintains a stable growth trend from 2012 to 2019, and the coupling coordination shows a spatially increasing distribution from upstream to downstream. From a temporal perspective, the city coupling coordination maintained an average annual growth of 3.42% from 2012-2019, from 0.4371 in 2012 to 0.7108 in 2019.From a spatial perspective, the main hierarchical types of urban coordination development levels in the Yangtze River Economic Belt changed from approaching imbalance and slight imbalance to primary coordination and intermediate coordination during 2012-2019. The coupling coordination of the Yangtze River Economic Belt shows an overall spatial distribution of gradient increase from the upstream region to the downstream region in 2012-2019. The spatial center of gravity shifts from (113°20'19 "E, 30°01'25 "N) to (113°20'19 "E, 30°01'25 "N). The spatial pattern is dominated by the northeast-southwest direction, and the long semi-axis of the standard deviation ellipse grows from 838.03 km to 845.99 km, and the short semi-axis grows from 316.81 km to 318.40 km. The overall dispersion level of the coupling coordination degree shows an overall trend of expansion. (3) The future spatial pattern of the coupling coordination degree of urban low-carbon economy efficiency and digital finance in China's Yangtze River Economic Belt will be dominated by the northeast-southwest direction and shift in the counterclockwise direction, and the dispersion level will show different trends in different directions. It is predicted that the spatial center of gravity of the coupling coordination of the two systems in the cities of the Yangtze River Economic Belt in China will be located in Jingzhou in the middle reaches of the Yangtze River during 2019-2040, and will move 22.17 km to the southwest. The rotation angle of the standard deviation ellipse will change from 71.26° in 2019 to 70.53° in 2040, and the long semi-axis of the standard deviation ellipse will decrease by 0.47% and the short semi-axis will increase by 2.12%. The spatial distribution of the coupling coordination will show an overall trend of dispersion in the northwest-southeast direction, while it will show an overall trend of insignificant concentration in the northeast-southwest direction. (4) The degree of coupling and coordination between urban low-carbon economy efficiency and digital finance in cities of the Yangtze River Economic Zone in China is influenced by various factors and its influence varies from period to period. In general, the driving factors that affect the coupling and coordination of the two systems in cities are: population development level, informatization level, greening level, transportation level, industrial structure, and research level. Among them, the influence of informatization level and industrial structure showed an overall increasing trend during the study period, with q-values increasing by 1.17% and 0.73%, respectively. The influence of population development level, greening level, transportation level, and scientific research level as a whole shows a gradual weakening, with q-values decreasing by 0.35%, 0.33%, 0.29%, and 1.70%, respectively, but the influence of population development level and scientific research level is still at a high level compared with other driving factors. 

5.We conclude the main results as follows: (1) in the time dimension, the coupling and coordination between urban low-carbon economy efficiency and digital finance increased by 3.42% annually, reflecting the increasingly coordinated development of the two systems; (2) in the spatial dimension, the coupling coordination between urban low-carbon economic efficiency and digital finance shows a distribution pattern of gradient increase from the upstream region of Yangtze River to the downstream region; (3) In terms of dynamic evolution, it is expected that the spatial center of gravity of the coupling and coordination between urban low-carbon economy efficiency and digital finance will move 22.17 km to the southwest from 2019 to 2040; (4) in terms of influencing factors, the effects of information technology level and industrial structure on the coupled coordination of urban low-carbon economic efficiency and digital finance increase over time, while the effects of factors such as population development level, greening level, transportation level, and research level weaken over time. 

Based on the summary of the above responses, we deleted the abbreviations of standard deviation ellipse and Global Malmquist-Luenberger in the original “Abstract” section, and added quantitative and specific results such as the annual average growth rate of coupling coordination and the distance of spatial center of gravity movement to the conclusion of the new “Abstract” section. The revised “Abstract” section is as follows:

China is a large country with rapid economic expansion and high energy consumption, which implies that the country's overall carbon emissions are enormous. It is vital to increase urban low-carbon economy efficiency (ULEE) to achieve sustainable development of China's urban economy. Digital finance is a significant tool to boost ULEE by providing a convenient and effective funding channel for urban low-carbon economic transformation. Analyzing the coupled and coordinated relationship between ULEE and digital finance is of vital importance for the sustainable development of the urban economy. This paper selects panel data of 100 cities in China's Yangtze River Economic Belt (YEB) in 2011-2019, and analyzes the research methods such as the Global Malmquist-Luenberger index model, coupling coordination degree (CCD) model, standard deviation ellipse model, gray model, and geographic detector by The spatial and temporal distribution, dynamic evolution characteristics and influencing factors of the CCD between ULEE and digital finance are analyzed. The study shows that : (1) the CCD of ULEE and digital finance grows by 3.42% annually, reflecting the increasingly coordinated development of the two systems; (2) The CCD of ULEE and digital finance shows a distribution pattern of gradient increase from the upstream region of Yangtze River to the downstream region, meanwhile, the spatial center of gravity moves mainly in the midstream region; (3) The spatial center of gravity of CCD of ULEE and digital finance is expected to move 22.17 km to the southwest from 2019 to 2040; (4) In terms of influencing factors, the influence of informatization and industrial structure on the CCD increases over time, while the influence of factors such as population development, greening, transportation, and scientific research decreases over time. Finally, this paper proposes policy recommendations for improving the CCD of ULEE and digital finance based on the empirical results.

2. The current Introduction should be further improved. A good one includes at least four aspects: motivation/background, literature review, aim and contribution, and organization of the remains of the study. Avoiding to put massive bibliographies behind one sentence. Such as XXXXX [1-5], OR 1, 2, 3, 4, 5; all references should be cited with detailed and specific descriptions.

Response: We found this comment is important and useful. We have improved the “Introduction” section around four areas: motivation/background, literature review, aim and contribution, and organization of the remains of the study. We have mainly made the following adjustments:

1.In terms of motivation/context, first, we describe the fact that China is a large carbon emitting country and the excessive carbon emissions in the Yangtze River Economic Zone of China, emphasize the harm caused by carbon emissions, and propose the necessity of improving low-carbon economy efficiency in cities. Then, we discuss the close relationship between digital finance and urban low-carbon economy efficiency. On the one hand, the development of digital finance has a supportive role in improving the efficiency of low-carbon economy in cities. Digital finance provides an efficient financing channel for low-carbon industries, and digital finance also reduces carbon emissions generated by offline financing activities. On the other hand, the efficiency improvement of urban low-carbon economy promotes the development of digital finance. Low-carbon economy efficiency improvement in cities accelerates industrial restructuring, which is beneficial to the development of digital finance industry. Urban low-carbon economy efficiency enhancement indicates an improved atmospheric environment, which is conducive to attracting excellent financial talents and technicians. Finally, the above discussion concludes that it is important to study the coupled and coordinated relationship between urban low-carbon economy efficiency and digital finance and its influencing factors.

2.For the literature review, we adjusted the literature citations in the “Introduction” section to add literature on the hazards of carbon emissions. Albouy’s (2020) research shows that global warming caused by carbon dioxide emissions can bring harm such as melting glaciers, rising sea levels, and decreasing species diversity (doi:10.1038/s41598-019-57280-3). Yu’s (2018) research shows that global warming will also increase the frequency of extreme weather events such as floods, droughts, hailstorms, tropical storms and tornadoes (doi:10.1016/j.cosust.2018.03.008). We checked the literature citations to ensure that we did not place a large amount of literature after a sentence. For a detailed presentation of the literature related to this paper, we describe in the “Literature review” section.

3.In terms of aims and contributions, we found that the current research mainly focuses on calculating and analyzing the low-carbon economy efficiency of each region, while there is a gap in the research on the coupled coordination relationship between urban low-carbon economy efficiency and digital finance. Meanwhile, the existing literature lacks the analysis of the future spatial pattern of the coupling coordination between urban low-carbon economy efficiency and digital finance, and the exploration of the influencing factors of the coupling coordination degree. In addition, the evaluation index system of low-carbon economy efficiency in the existing literature considers a single perspective and needs to be improved. In order to fill the above research gaps, we propose three issues that are the focus of this paper: (1) How has the urban low-carbon economy efficiency in the Yangtze River Economic Belt of China changed over the past few years? (2) How has the coupling and coordination relationship between the urban low-carbon economy efficiency and digital finance in China's Yangtze River Economic Belt changed over the past few years, and how will the spatial pattern change in the future? (3) What are the factors affecting the coordinated development of urban low-carbon economy efficiency and digital finance in the Yangtze River Economic Belt of China? We believe that the most important contribution of this study is that it provides, for the first time, a reliable model for evaluating the level of coordinated development of urban low-carbon economy efficiency and digital finance. Meanwhile, we analyze for the first time the spatial and temporal evolution characteristics, future spatial patterns, and influencing factors of the coupled coordination relationship between urban low-carbon economy efficiency system and digital finance system. In addition, our study improves the evaluation framework of low-carbon economy efficiency from the perspective of public finance and land use. This study provides a new path to promote the coordinated development of low-carbon economy efficiency and digital finance in the Yangtze River Economic Belt region, which helps the government to formulate environmental protection policies, accelerate the low-carbon transformation of cities, and promote sustainable development.

4.In terms of the organization of the research remains, the “Literature review” section introduces the literature related to this paper, and the “Materials and methods” section describes the construction of the evaluation index system, the selection of the sample, the data sources and the research methods used in this study. The “Results” section answers the three questions raised in the introduction section. The first part of this section analyzes the spatial and temporal evolution characteristics of urban low carbon economy efficiency, the second part analyzes the spatial and temporal evolution characteristics of the coupling coordination degree of urban low carbon economy efficiency and digital finance and the future spatial pattern changes, and the third part analyzes the changes of the influencing factors of the coupling coordination degree. The last section summarizes the research findings and puts forward relevant policy recommendations.

We have improved the “Introduction” section based on the above four aspects, and the revised “Introduction” section is as follows:

It is well known that global warming due to the continued growth of CO2 emissions has not only brought about glacial melting, sea-level rise, and reduction in species diversity (Albouy, 2020). It also increases the frequency of extreme weather events such as floods, droughts, hailstorms, tropical storms, and tornadoes (Yu et al., 2018). China is one of the countries with the highest total carbon emissions in the world (Rovinskaya, 2020). In 2016, China's carbon emissions were 1.97 times higher than those of the United States, accounting for 27.49% of the world's total emissions. As one of the most important urban economic zones in China, the Yangtze River Economic Belt (YEB) not only spans the eastern, central, and western regions of China but also plays a vital role in China's economic development with a population size of 600 million people and a dense urban distribution (Li et al., 2021). In 2018, China's YEB region contributed 44.1% of China's GDP (Pan et al., 2020), but also produced the most carbon emissions . The reason for this is that the YEB gathers many key national manufacturing projects such as steel, petrochemicals, automobiles, and electromechanics (Tang et al., 2019), and these manufacturing projects consume a large amount of fossil energy, which is what causes the total CO2 emissions of the YEB to be too large (Yi et al., 2020). In 2013, for example, the carbon emissions of YEB accounted for about 44.6% of the national carbon emissions. Some scholars point out that developing a low-carbon economy is an efficient strategy to deal with high energy usage and carbon emissions (Tsai et al., 2016). Low carbon economics states that urban low-carbon economy efficiency (ULEE) can be seen as an environmental efficiency that measures the ability of an urban economic development process to increase regional GDP with fewer factor inputs and produce fewer carbon emissions (Zhang et al., 2020). Therefore, improving the ULEE of YEB is an important way to alleviate environmental pressure in China.

With the fast growth of digital finance, the connection between ULEE and digital finance has become closer. On the one hand, digital finance contributes the growth of the low-carbon economy. The advancement of digital finance improves the efficiency of financing (Liu et al., 2016), which can provide sufficient financial support for low-carbon industries, help low-carbon industries develop rapidly, encourage the improvement of industrial structure, and effectively reduce carbon emissions, thus boosting the growth of ULEE. At the same time, digital finance can also help high-carbon emission enterprises upgrade their energy utilization technologies through financial support (Xie et al., 2020), thus realizing the growth of the urban low-carbon economy. In addition, digital finance is convenient and inclusive (Zhang et al., 2020), and digital financial services can be completed through cell phones, personal computers, the Internet, or with reliable digital payment systems (Ozili, 2021), which can reduce the carbon emissions generated by SMEs and individual consumers traveling offline to and from financial institutions to participate in financial services. On the other hand, the improvement of ULEE can foster the healthy growth of digital finance. The improvement of ULEE means that the relevant resource elements in the region are better allocated and applied (Ozili, 2018), which can provide a better soft and hard environment for the digital finance industry's growth.. The growth of the low-carbon economy accelerates the transformation of industrial structure and accelerates the development of high-tech industries with low-carbon row characteristics, which will provide a higher level of digital technology support for digital finance. In addition, the improvement of ULEE indicates that the urban atmosphere is improved, which can improve the attractiveness of the city to financial and high-tech talents (Zhao et al., 2017), thus enabling the growth of local digital finance. In summary, analyzing the development relationship between ULEE and digital finance in the YEB region and the influencing factors is of non-negligible relevance to the low-carbon transition and sustainable development of Chinese cities.

According to existing research, this paper finds that the current literature mainly focuses on calculating and analyzing the efficiency of low carbon economy in each region (Tan et al., 2017), while there is a gap in the research on the coupled coordination relationship between ULEE and digital finance. Meanwhile, the existing literature lacks the analysis of the future spatial pattern of coupling coordination degree (CCD) between ULEE and digital finance and the exploration of the influencing factors of CCD. In addition, the assessment indicator system of low carbon economy efficiency in the existing literature considers a single perspective and needs to be improved (Liu et al., 2019). Therefore, to remedy these shortcomings, this paper focuses on addressing three issues regarding the developmental relationship between ULEE and digital finance in the Chinese YEB region:

·How has the ULEE in the YEB region of China changed over the past few years?

·How has the coupling and coordination relationship between ULEE and digital finance in China's YEB region changed over the past few years, and how will the spatial pattern change in the future?

·What are the factors affecting the coordinated development of ULEE and digital finance in the YEB region of China?

These questions are closely related to urban low-carbon transition as well as sustainable development. The most important contribution of this study is to provide the first reliable model for evaluating the level of coordinated development of ULEE and digital finance, using 100 cities in the YEB region of China as examples. Based on the standard deviation ellipse model, gray model, and geographic detector, the spatial and temporal evolution characteristics, future spatial patterns, and impact factors of the coupled and coordinated relationship between the ULEE system and digital financial system are analyzed for the first time. In addition, this study improves the evaluation framework of low carbon economy efficiency from the viewpoint of public finance and land usage. This research provides a new path to foster the coordinated development of low-carbon economy efficiency and digital finance in the YEB region, which helps the government to formulate environmental protection policies, accelerate the low-carbon urban transformation, and boost sustainable development.

For the rest of the paper, “Literature review” section describes the literature related to this paper. “Materials and methods” section elaborates on the construction of the indicator system, sample selection, data sources, and models used in this study. “Results” section is the analysis of the empirical results and discusses three issues.The first part of this section analyzes the spatial and temporal evolutionary characteristics of ULEE, the second part of this section analyzes the spatial and temporal evolutionary characteristics and future spatial patterns of CCD, and the third part of this section detects the influencing factors of CCD. Conclusions and policy recommendations are presented in “Conclusion and countermeasures” section.

3. Literature Review has the chance to be further improved: it seems that the authors have made the retrospection. However, via the review, what issues should be addressed? What is the current specific knowledge gap? What implication can be referred? The above questions should be point-by-point answered clearly. This work focused on the low carbon transition in economic growth in 100 cities. At the consumption side, buildings show the most significant potential in cost-effectiove emission reduction, which is worthy to be discussed. Here, some latest literature investigating the low carbon roadmap or carbon neutral pathway of building sector is worhy to added and discussed. E.g., DOI: 10.1016/j.eneco.2021.105712 DOI: 10.1016/j.apenergy.2021.118098 DOI: 10.3390/buildings12010054 DOI: 10.3390/buildings12010083 DOI: 10.3390/buildings12020128

Response: Thank you very much for your valuable comments, and we have made the following improvements to the literature review section in response to your questions and suggestions:

1.Appriciate you very much for the recommended literatures. We suppose that these literatures are valuable and could give a deeper insight to carbon reduction in the construction industry. As recommended, we have cited these studies (Sun et al., 2022, doi: 10.3390/buildings12020128; Zhang et al., 2022, doi: 10.1016/j.eneco.2021.105712; Li et al., doi: 10.1016/j.apenergy.2021.118098; Xiang et al., 2022, doi: 10.3390/buildings12010054; Xiang et al., 2022, doi: 10.3390/buildings12010083) on lines 110 through 127 of the ‘Manuscript'.

2.By reviewing the literature, we have identified three problems that need to be addressed in this paper: (1) How has the urban low-carbon economy efficiency in the Yangtze River Economic Belt of China changed over the past few years? (2) How has the coupling and coordination relationship between the urban low-carbon economy efficiency and digital finance in China's Yangtze River Economic Belt changed over the past few years, and how will the spatial pattern change in the future? (3) What are the factors affecting the coordinated development of urban low-carbon economy efficiency and digital finance in the Yangtze River Economic Belt of China? The above three problems we have presented in the “Introduction” section, so they are not repeated in the “literature review” section. We focus on highlighting the current knowledge gaps in the “Literature review” section.

3.By combing through the literature, we conclude that the current knowledge gaps in the research on the coupled and coordinated relationship between low-carbon economy efficiency and digital finance in cities are mainly 2 points: (1) the current low-carbon economy efficiency assessment index system is constructed from a single perspective and needs to be improved. For most studies on measuring low-carbon economy efficiency, the assessment framework is mainly constructed around capital, labor, energy, GDP, and carbon emissions, ignoring the role of public finance and land use on the development of low-carbon economy. (2) So far, there are no suitable indicators to measure the changes in the coupled and coordinated relationship between low carbon economy efficiency and digital finance. We have done the following work to narrow the above knowledge gap: (1) We have improved the evaluation index system of urban low-carbon economic efficiency from the perspective of public finance and land use. A new urban low carbon economic efficiency evaluation index system was established from seven aspects of capital, labor, public finance, land, energy, GDP and carbon emission, and the GML index model was applied to measure the urban low carbon economic efficiency of 100 cities in the Yangtze River Economic Belt region of China. (2) The coupling coordination between ULEE and digital finance in 100 cities in China's Yangtze River Economic Belt from 2012 to 2019 is measured, and its influencing factors are analyzed. This paper selects the Digital Inclusive Finance Index of Peking University and the measured urban low-carbon economic efficiency to measure the coupling coordination degree of China's Yangtze River Economic Belt using the coupling coordination degree model, and further analyzes the spatial and temporal evolution characteristics, future spatial characteristics and influencing factors of the coupling coordination degree.

4.After combing through the literature, we summarize the ways of obtaining carbon emission data in cities, the relationship between low carbon economy and finance in cities, and the methods used to study the coupled and coordinated relationship between low carbon economy efficiency and digital finance in cities. In the first part of the literature review, we reviewed the relevant literature and found that the official direct release of carbon emission data in China is limited, and we need to obtain carbon emission data through measurement. Carbon emission measurement based on consumption measurement can effectively avoid the underestimation and overestimation of carbon emissions caused by trade. In the second part of the literature review, we can learn from the literature that there is a strong relationship between low carbon economy and finance. As an innovation of traditional financial services and products based on digital technology, there is a complex relationship between digital finance and low-carbon economy that both promotes and constrains each other. In the third part of the literature review, we clarify that the coupled coordination degree model is a model that is widely used to study the coupled coordination relationship between multiple systems at present.

After the above improvements, our “Literature review” section is as follows:

The reliability of the research data ensures the credibility of the research conclusions. Currently, the official regional carbon emission data published in China are very limited, and the regional carbon emission data are mainly obtained through measurement. According to the existing literature, regional carbon emission measurement is mainly based on two methods: production-based accounting and consumption-based accounting (Shao et al., 2016). Many scholars have measured CO2 emissions using production-side-based accounting and conducted related studies. Shan (2018) measured energy-related carbon emissions and industrial process-related carbon emissions in China from 1997 to 2015 from a production perspective regarding IPCC guidelines. Wang (2020) measured carbon emissions in China and India using a production-based perspective and compared the dynamic evolution and drivers of the two. Some scholars argue that measuring carbon emissions based on the production perspective will ignore the transfer of emissions due to import trade and create a "carbon leakage" phenomenon (Jakob et al., 2014). Bai (2021) measures consumption-based carbon emissions in the Beijing-Tianjin-Hebei region of China and analyzes the differences and drivers of emissions between cities from 2012 to 2015 . Qian (2022) measured the carbon emissions of 47 cities in the Pearl River Basin of China using a consumption-based carbon emissions accounting approach and found that 47 cities accounted for 13.1% of China's emissions and that there were large differences in carbon emissions between cities . It is worth mentioning that, on the consumption side, the building industry has great potential and research value in reducing carbon emissions, which has been analyzed and discussed by several researchers. Sun (2022) used bibliometric methods to analyze and summarize 364 articles published from 1990 to 2021 on peak carbon and carbon neutrality in the building sector. Zhang (2022) evaluated the carbon emission reduction and carbon emission reduction efficiency of commercial buildings in China and the U.S. at different scales and mapped the energy efficiency improvement paths of commercial buildings in both countries. Li (2022) established an assessment framework of emission reduction intensity, quantity, and efficiency through carbon intensity decomposition and evaluated the carbon emission reduction changes of commercial buildings in 30 provinces in China from 2001 to 2016. Xiang (2021) measured the carbon emissions of commercial buildings in China and then estimated them using LASSO regression, optimized the nonlinear parameters using a whale optimization algorithm, and found that the peak emissions in the commercial building sector were 1264.81 MtCO2, and the main drivers of carbon emissions were population size and energy intensity. Xiang (2022) developed a novel open-source tool PyLMDI based on the LMDI method and used it to analyze the carbon reduction potential of commercial buildings in China and the United States.

Digital finance refers to all comprehensive applications that rely on digital technology to innovate traditional financial products and service forms. Its essence is to empower traditional finance to solve the problem of high risk and high cost arising from information asymmetry by using modern information technology such as artificial intelligence, cloud computing, blockchain, and big data, to change the link of value delivery in the traditional financial model, and to provide richer financial products while reshaping the traditional financial system. With the rapid growth of the low-carbon economy, an increasing number of scholars have started to take notice of the relation between the low-carbon economy and finance. Some scholars believe that finance can help the low-carbon economy flourish (Qin et al., 2021). On the one hand, finance can control carbon emissions through the carbon trading market under the market mechanism (Qi etal., 2021). Wang (2019) studied the effectiveness of the carbon emissions trading pilot in China from 2007 to 2017 based on the robust regression algorithm of M estimation and found that the carbon emissions intensity in China is diminishing with each passing year, and the carbon emissions intensity is lower in regions with higher economic level, and the carbon emissions trading pilot has a significant driving effect on reducing carbon emissions . Guo (2021) examined the impact of carbon emissions trading policies on the financing of carbon emission reduction and carbon emissions in China, and the findings revealed that carbon emissions trading policies can effectively promote the financing of carbon emission reduction and reduce carbon emissions in China, with more significant impacts in the eastern and affluent areas, and the impacts are persistent. On the other hand, finance can build links with the real economy by providing financing for low-carbon projects (Zahoor et al., 2021), bank loans (Umar et al., 2021), etc., so that the carbon emissions of emitters can be effectively reduced. Paroussos (2019) uses a large-scale application of the CGE model in the context of global GHG emission reduction to measure the macroeconomic impact of investments required to reduce the GHG emissions generated by the Italian energy system by 76% compared to 1990 levels. From the results, it is clear that low-cost financial resources and the availability of market share and rapidly advancing clean energy technologies would benefit Italy in its low-carbon economic transition. Schumacher (2020) investigates the role of sustainable finance in supporting Japan's transition to a zero-carbon, sustainable economy, as well as the impact of policies and regulations in increasing investment in sustainable finance and low-carbon infrastructure. The findings show that the Japanese financial sector should increase the integration of sustainable finance and ESG principles across all asset classes in its investment portfolio in order to finance a net-zero carbon economy. Sartzetakis (2020) analyzed the important role of green bonds in the transformation of the low-carbon development approach based on the theory of intergenerational burden and the need for large long-term infrastructure construction. Sun (2021) constructed a neural network-based correlation analysis model between green finance and carbon emissions and conducted simulation tests to check the validity of the results, and found that there is a significant correlation between green finance and carbon emissions. Elheddad (2020) studied the effect of e-finance on carbon emissions by selecting panel data for 29 OECD countries from 2007 to 2016, controlling for possible heterogeneity between countries using fixed and random effects models, and testing robustness using instrumental variables estimation methods and panel quantile regressions, which showed that the development of e-finance mitigates carbon emissions in OECD countries and plays an important role in environmental protection. In addition, a low-carbon economy also has an essential role in financial development, as the growth of a low-carbon economy cannot be separated from the strengthening of low-emission infrastructure, the vigorous development of clean energy, and the upgrading of industrial structures, and the development of these activities will indirectly promote the development of the financial sector (Genget al., 2020). Obviously, digital finance, as an innovation of traditional forms of financial services and financial products relying on digital technology, has a complex relationship with the low-carbon economy that both promotes and constrains each other.

Coupling coordination degree (CCD) is a method to analyze the correlation relationship between multiple systems, which can effectively reflect whether the relationship between multiple systems is harmonious and well-matched, with an overall trend of coordinated development (Wang et al., 2019). At present, some researchers have applied the CCD model to investigate the coupled coordination relationship between different industries and carbon emissions. Han (2018) analyzed the CCD between agricultural carbon emissions and the agricultural economy in 30 Chinese provinces from 1997 to 2015 and studied the potential drivers using the LDMI decomposition model. Pan (2021) used the SBM-DEA model combined with the CCD model to measure the coupled coordination of carbon emissions, economic development, and regional innovation in tourism and analyzed the core influencing factors using the geographic detector. In addition, some scholars have studied the coupled coordination relationship between the carbon emission system and other systems by constructing a CCD model. Shen (2018) measured the CCD between socio-economic and carbon emissions using an improved CCD model by selecting data from 30 Chinese provinces. Song (2018) measured the coupled coordination relationship between carbon emissions and urbanization using the coordination degree model and CCD model, respectively, based on data from 30 Chinese provinces. Chen (2020) studied the degree of coordination between carbon emissions and the ecological environment in China from 2009 to 2015 using the CCD model and used the log-mean divisor exponential decomposition method to determine the key factors affecting the degree of coordination. Zhou (2020) analyzed the CCD between carbon emission efficiency and industrial structure improvement in each province of China and designed the coupling paths using a distributional dynamics framework.

The above describes the data acquisition, the relationship between finance and low-carbon economy, and the choice of research methods. A review of the literature reveals that there are two specific knowledge gaps regarding the developing relationship between urban low-carbon economy efficiency (ULEE) and digital finance.

·The current low carbon economy efficiency assessment index system is constructed from a single perspective and needs to be improved.

For most studies on measuring low-carbon economy efficiency, the assessment framework is mainly constructed around the capital, labor, energy, GDP, and carbon emissions (Zhang et al., 2017), ignoring the role of public finance and land use in the advance of the low-carbon economy. Public finance can effectively stimulate low-carbon innovation (Owen et al., 2018), which has an important role in the low-carbon transition of cities. Meanwhile, land use is the second-largest source of carbon emissions after fossil fuels (Huang et al., 2015). Considering the importance of public finance and land use to the growth of the urban low-carbon economy, this paper includes them in the evaluation index system of ULEE.

·Till now, there is no suitable indicator to measure the change in the coupled and coordinated relationship between ULEE and digital finance.

Traditional finance has undergone a digital transition, and digital finance is a new manifestation of that development. Digital finance can provide users with more convenient and efficient financial services with the help of the Internet and digital technology, and significantly increase the efficiency of financing for individuals and enterprises (Hu et al., 2016). The existing researches have mainly concentrated on analyzing the relationship between low-carbon economy and traditional finance (Zhang, 2011), and there are gaps in the research on the relationship between ULEE and digital finance, especially the research on the coupling and coordination relationship between the two systems. The coupling and coordination relationship between ULEE and digital finance lacks theoretical mechanisms and suitable measurement indicators. In addition, the influencing factors and future spatial pattern prediction of the CCD between the two systems also need to be studied. The study of the CCD between ULEE and digital finance and the influencing factors will help the coordinated development of the low-carbon economy and digital finance, accelerate the low-carbon transformation of Chinese cities, promote the process of global CO2 emission reduction, and enhance sustainable development.

Accordingly, this study endeavors to narrow these gaps through the following efforts.

·Improved the assessment indicator system of ULEE from the perspective of public finance and land use.

This research proposes a new evaluation index system of ULEE from seven aspects: capital, labor, public finance, land, energy, GDP, and carbon emission, and measures the ULEE of 100 cities in China's Yangtze River Economic Belt (YEB) region using the GML index model. Meanwhile, this paper analyzed the changes of ULEE from the time perspective, analyzed the distribution differences of ULEE from the spatial perspective, and decomposed the ULEE index to explore the reasons for the changes in ULEE.

·Measured the CCD of ULEE and digital finance in 100 cities in the YEB region of China from 2012 to 2019, and analyzed its influencing factors.

This paper measures the coupled coordination relationship between ULEE and digital finance in the YEB region for the first time by combining ULEE and Peking University Digital Inclusive Finance Index using the CCD model and conducts a spatio-temporal evolution analysis with the Standard deviation ellipse (SDE) model to discuss the changes in the CCD of the two systems and the reasons. Meanwhile, this paper also conducts a time series prediction of the parameters of SDE based on the gray model and analyzes for the first time the changes in the spatial distribution of the CCD of ULEE and digital finance in 2019-2040. In addition, to further promote the CCD of ULEE and digital finance and sustainable urban development, this paper detects for the first time the changes in the influence of informatization, industrial structure, population development, greening, transportation, and scientific research on CCD between ULEE and digital finance by using geographic detectors.

4. For some details, be sparing in the use of tables and ensure that the data presented in them do not duplicate results described elsewhere in the article. It is suggest to avoid using vertical rules and shading in table cells. It is also suggested that the key findings should be summarized in Conclusion one by one with the marks of 1. 2. 3. or i. ii. iii. etc.

Response:Thank you for your reminders and suggestions, we have checked and improveed the details you raised. We deleted Table 5 from the original draft and used text to express the contents of the table for the purpose of saving tables. 

Table5.Reliability test of prediction results

variables average relative error(%) average posterior difference ratio C average probability of small error P development gray value a Relevance value r

Parameters of SDE 0.03615 0.1875 0.95 0.0002 0.6200

We expressed the above table by converting it into the following text: the average relative error of the prediction results was 0.0362%, which was less than 1%. The average posterior difference ratio is 0.1875, which is less than 0.65. the average small error probability is 0.95, which is greater than 0.7. The development gray value is 0.0002, which is less than 0.3, and the correlation coefficient is 0.62, which is greater than 0.6. The above test results indicate that the grey model accuracy test is passed, and the prediction results are good and credible. 

We also checked the rest of the tables in the article to make sure that the table content was not duplicated elsewhere in the article and that the tables did not have formatting issues such as shading. We summarized the results of the analysis and labeled them with numbers in the “Conclusion” section., as follows:

1.The urban low-carbon economy efficiency (ULEE) of China's Yangtze River Economic Belt (YEB) region shows a slow fluctuating upward trend from 2012 to 2019, and the ULEE shows a spatial distribution of the strongest in the upstream region, the second strongest in the downstream region, and the weakest in the midstream region. From a temporal perspective, the ULEE in China's YEB maintains an average annual growth of 0.93% during the study period. After the decomposition of indicators, it can be seen that pure technical efficiency (PEC) and scale efficiency (SEC) grew by 13.16% and 4.80%, respectively, during the study period, which drove the growth of ULEE. From a spatial perspective, ULEE has variability in spatial distribution, with 74 cities out of 100 having urban low carbon economic efficiency values above 1. Among them, Yuxi, Jingzhou, Zunyi, Shanghai, Lijiang, Changzhou, Qujing, and Guangyuan have relatively high ULEE. The other 26 cities still have more room for improvement in terms of economic development and carbon emission reduction.

2.The Coupling coordination degree (CCD) of ULEE and digital finance in China's YEB region maintained a stable growth trend from 2012 to 2019, and the CCD showed a spatially increasing distribution from upstream to downstream. From a temporal perspective, urban CCD maintained an average annual growth of 3.42% from 2012 to 2019, from 0.4371 in 2012 to 0.7108 in 2019. From a spatial perspective, the main hierarchical types of urban CCD in the YEB region changed from approaching imbalance and slight imbalance to primary coordination and intermediate coordination from 2012 to 2019. The CCD in the YEB region shows an overall spatial distribution of gradient from the upstream region to the downstream region from 2012 to 2019. The spatial center of gravity shifts from (113°20'19 "E, 30°01'25 "N) to (113°20'19 "E, 30°01'25 "N), and the spatial pattern is dominated by the northeast-southwest direction, with the long semi-axis of SDE growing from 838.03km to 845.99km and the short semi-axis growing from 316.81km to 318.40km. The overall level of dispersion of the CCD shows an overall trend of expansion.

3.The future spatial pattern of the CCD of ULEE and digital finance in the YEB region of China will be dominated by the northeast-southwest direction and shift in the counterclockwise direction, and the dispersion level will show different trends in different directions. It is predicted that during 2019-2040, the spatial center of gravity of the CCD of the two major urban systems in the YEB region of China will be located in Jingzhou in the midstream of the Yangtze River region, and the overall will move 22.17 km to the southwest. The rotation angle of Standard deviation ellipse (SDE) will change from 71.26° in 2019 to 70.53° in 2040, the long semi-axis of SDE will decrease by 0.47% and the short semi-axis will increase by 2.12%. The spatial distribution of CCD will show an overall trend of dispersion in the northwest-southeast direction, and an overall trend of insignificant concentration in the northeast-southwest direction.

4.The CCD of ULEE and digital finance in the YEB region of China is influenced by various factors and their influence varies over time. In general, the driving factors affecting the CCD of the two urban systems are population development level, informatization level, greening level, transportation level, industrial structure, and research level. Among them, the influence of informatization level and industrial structure showed an overall trend of enhancement during the study period, with q-values increasing by 1.17% and 0.73%, respectively. The influence of population development level, greening level, transportation level, and scientific research level as a whole shows a gradual weakening, with q-values decreasing by 0.35%, 0.33%, 0.29%, and 1.70%, respectively, but the influence of population development level and scientific research level is still at a high level compared with other driving factors.

5. As a key part of a paper, Discussion should show the readers at least two elements: "breadth" and "depth". "Breadth" reflects whether the analytical results can be explained via different approaches. "Depth" reflects whether the analytical results completely answer the questions raised in Introduction. My first sense shows the current Discussion is without enough insight. This should explore the significance of the results of the work, not repeat them. A combined Results and Discussion section is OK. However, avoid extensive citations and discussion of published literature.

Response:We appreciate for your comments and we have followed your suggestions to improve the “Results” section of the article both in terms of “breadth” and “depth”.

1.In terms of “breadth”, we analyzed the changing characteristics of urban low carbon economy efficiency in two dimensions: time and space, respectively. Temporally, from 2012 to 2019, the urban low-carbon economy efficiency of the Yangtze River economic belt in China showed a fluctuating upward trend with an average annual growth of 0.93%. Spatially, the mean values of low-carbon economy efficiency of cities in the upper, middle and lower Yangtze River region from 2012 to 2019 are 1.0172, 1.0062 and 1.0077, respectively, all of which exceed 1. The performance of urban low-carbon economy efficiency in China's Yangtze River Economic Belt in both dimensions indicates that the region has performed well overall in terms of low-carbon economy efficiency from 2012-2019, and the low-carbon economy has been developed effectively. We also analyze the changes in the coupling coordination between urban low-carbon economy efficiency and digital finance in the Yangtze River Economic Belt in both temporal and spatial dimensions. Temporally, the coupling coordination degree (CCD) showed an overall upward trend from 2012-2019, with an average annual increase of 3.42% in the coupling coordination. Spatially, the grade type of the coupling coordination degree of most cities in the Yangtze River Economic Belt is approaching imbalance (0.40<CCD<0.49) and slight imbalance (0.30<CCD<0.39) in 2012, accounting for 72% of the total number of cities studied. And the grade type of coupling coordination degree in most cities in 2019 is intermediate coordination (0.70<CCD<0.79) and primary coordination (0.60<CCD<0.69), accounting for 97% of the total number of cities studied. The change characteristics of the coupling coordination degree of urban low-carbon economy efficiency and digital finance in both time and space indicate that the level of coordination development of urban low-carbon economy efficiency and digital finance has improved significantly in 2012-2019 as a whole, and the conclusions of the analysis of spatial and temporal characteristics are consistent.

2.In terms of “depth”, we provide a more detailed analysis of the spatial and temporal evolutionary characteristics of urban low-carbon economy efficiency and coupled coordination degree, and further discuss the reasons for the analysis results in order to be able to fully answer the questions raised in the “Introduction” section.

(1)In the “Analysis of ULEE (urban low-carbon economy efficiency)” section, we add that the overall growth trend in low-carbon economy efficiency of cities in China's Yangtze River Economic Zone from 2012 to 2019 is due to a series of policies formulated by the Chinese government during this period to promote the development of a low-carbon economy, encouraging enterprises to use clean energy and improve energy use efficiency, significantly improving pure technical efficiency, driving the growth of low-carbon economy efficiency. We also add the specific changes in low-carbon economy efficiency of cities in the Yangtze River Economic Zone for each year during 2012-2019. Among them, the urban low-carbon economy efficiency increased by 1.4% in 2015, mainly because the Chinese government strengthened its financial support for low-carbon industries and further implemented tax relief policies in 2015, which promoted low-carbon technology innovation of enterprises. In addition, in 2015, the government increased the elimination of backward production capacity, adjusted and optimized the urban energy consumption structure, reduced the proportion of coal consumption, and encouraged the development of new clean energy, thus accelerating the development of low-carbon economy. After 2017, cities started to show a sequential increase in low-carbon economy efficiency, increasing by 4.41% and 4.07% in 2018 and 2019, respectively. This could be due to the official launch of the carbon emission trading market at the end of 2017. Under the market mechanism, market players with excess or insufficient allowances can accomplish their emission reduction targets more effectively through trading. The carbon trading mechanism reduces the cost of carbon reduction for enterprises, accelerates low-carbon technology innovation and promotion, and at the same time promotes the coordinated development of industrial structure, thus realizing the efficiency improvement of low-carbon economy. We also add the spatial distribution characteristics of urban low-carbon economy efficiency and its decomposition indicators in the Yangtze River Economic Belt. In general, the low-carbon economy efficiency in China's Yangtze River Economic Belt shows the strongest distribution in the upstream region, followed by the downstream region, and the weakest in the midstream region in space. After the decomposition of urban low-carbon economy efficiency, it is found that the decomposition indexes show different characteristics of spatial distribution in the Yangtze River Economic Belt respectively. Among them, the spatial distribution of pure technical efficiency is similar to that of urban low-carbon economy efficiency, with the upstream region having the relatively highest level of pure technical efficiency with a value of 1.0198, the midstream region having the relatively lowest level of pure technical efficiency with a value of 1.0118, and the downstream region having a level of pure technical efficiency between the upstream and midstream regions with a value of 1.0155. The spatial distribution of scale efficiency in the Yangtze River Economic Zone is mainly characterized by the decreasing scale efficiency from the upstream region to the downstream region, with 1.0110, 1.0050, and 1.0030 for the upstream, middle, and downstream regions, respectively. The spatial distribution of technological progress in the Yangtze River Economic Zone, on the other hand, is characterized by the strongest in the downstream region, followed by the upstream region and the weakest in the midstream region, with 0.9978, 0.9946 and 0.9934 in the downstream, upstream and midstream regions, respectively. From the spatial distribution of the decomposition indexes, it can be seen that the upstream region has the highest urban low-carbon economy efficiency mainly due to the efficient application of technology and the scale dividend, while the region needs to improve in terms of technological innovation. The low-carbon economy in the midstream region is in the scale development stage, and should expand production inputs and accelerate technological innovation to further improve the efficiency of the urban low-carbon economy, while the pure technical efficiency level in this region is the lowest among the three regions, and the region should strengthen the application and promotion of low-carbon technologies. The application level of low-carbon technologies in the downstream region is relatively high, and the region outperforms other regions in terms of technological innovation, but the strength of technological R&D is still insufficient. Meanwhile, the level of scale efficiency in the downstream region is lower than that in the upstream and midstream regions, and the scale dividend generated by expanding resource input in this region is relatively the least.

The above is an improvement and addition to our “Analysis of ULEE” section. “Analysis of ULEE” section mainly answers the first question raised in the introduction section: how has the urban low-carbon economic efficiency in China's Yangtze River Economic Belt changed over the past few years?

(2)In the “Spatial and temporal evolution of the CCD (Coupling coordination degree)” section, we add the specific temporal characteristics of the coupling coordination degree from 2012-2019 and analyze the reasons for this characteristic. From 2012 to 2016, the coupling coordination degree has been increasing annually, from 0.44 to 0.65. However, the value of coupling coordination degree is not high and the growth rate decreases year by year. This is mainly because digital finance is still in the early stage of development and the intensity of support for the development of low carbon economy is insufficient, resulting in the slow growth of urban low carbon economy efficiency, which slows down the coordinated development of urban low carbon economy efficiency and digital finance. The coupling coordination degree showed a small drop to 0.64 in 2017, which is related to the decline of urban low-carbon economy efficiency due to insufficient technological innovation, limiting the synergistic development of urban low-carbon economy efficiency and digital finance. After 2017, the coupling coordination degree has seen continuous growth, increasing to 0.71 in 2019. This is due to the fact that the launch of China's carbon emission rights market has led to effective control of urban carbon emissions, while the opening of the carbon emission rights market has stimulated industrial structure optimization and energy consumption restructuring, accelerating the development of low-carbon industries, which is conducive to promoting the development of digital finance industry, while the development of digital finance provides more financing opportunities for urban low-carbon economy transformation. Based on the above analysis, we can learn that China's Yangtze River Economic Belt city cluster has achieved remarkable results in promoting the synergistic development of urban low-carbon economy efficiency and digital finance during 2012-2019, and the level of coordination and development of the two systems is developing towards a better direction. We also add the reasons for the spatial distribution characteristics of the coupling coordination of the Yangtze River Economic Belt in China at four time points: 2012, 2014, 2016, and 2019. The overall level of coupling coordination of cities in the Yangtze River Economic Belt was relatively low in 2012. The cities with approaching imbalance and slight imbalance are mainly concentrated in the upstream region of Yangtze River, and the cities with reluctant coordination are mainly concentrated in the midstream and downstream regions of Yangtze River. This spatial distribution characteristic may be due to the fact that cities in the upstream region are mostly inland cities, where labor, technology, capital, and other resources are difficult to obtain, and production and operation are mostly based on rough industrial production, making it difficult to coordinate the development of urban economy and urban environment. Meanwhile, the development of the digital finance industry in the region is relatively slow and the financing support is insufficient, and the low-carbon transformation of cities is constrained by the sluggish development of digital finance, forcing the two to achieve coordination with increased difficulty. Compared with 2012, the overall level of coupling coordination of cities in the Yangtze River Economic Belt improved in 2014, except for Chengdu, Zunyi and Zhaotong in the upper Yangtze River region, which are still in a state of approaching imbalance, most of the cities approaching imbalance transformed into a state of reluctant coordination or slight coordination, mainly concentrated in the middle and lower reaches of the Yangtze River region. This may be because the middle and lower reaches of the Yangtze River have developed economies, advanced energy conservation and emission reduction technologies, reasonable energy consumption structures and relatively high levels of digital finance development, and digital finance can play a better financing role in the process of urban low-carbon economic development, so the development relationship between urban low-carbon economy efficiency and digital finance in the middle and lower reaches is more coordinated. In 2016, the coupling coordination of cities in the Yangtze River Economic Belt further improved, and some slight coordination cities transformed to intermediate coordination, such as Suzhou, Zhenjiang, Quzhou and Nanchang, mainly concentrated in the middle and lower reaches of the Yangtze River region. The reluctant coordination cities are mainly distributed in the upstream region of the Yangtze River, while the slight coordination cities show an even distribution throughout the Yangtze River Economic Belt, with a denser distribution in the middle and lower reaches of the Yangtze River. This may be due to the fact that as the level of digital finance development in the Yangtze River Economic Belt region increases, the financing channels for each city to carry out low-carbon transformation are expanded, low-carbon technology R&D and industrial restructuring are better supported, and the coupled and coordinated relationship between digital finance and urban low-carbon economy efficiency is optimized. In 2019, most cities in the Yangtze River Economic Belt are already in intermediate coordination, and only a small number of slightly coordinated cities are located in the upper and middle reaches of the Yangtze River region. This implies that the coordination relationship between low-carbon economy efficiency and digital finance in cities in the Yangtze River Economic Belt region is good overall, and there is room for improvement in a few cities in the upper and middle reaches of the Yangtze River region. In summary, the coordination between low-carbon economy efficiency and digital finance in cities in the lower Yangtze River region has the highest level of development during the study period, mainly due to the high level of economic development, advanced clean technology, reasonable energy consumption structure and relatively good development of digital finance in the region. The upper Yangtze River region has the lowest level of coordinated development of urban low-carbon economic efficiency and digital finance due to the harsh geographical conditions, backward production methods and relatively late start of digital finance. The level of coordinated development of the two systems in the middle reaches of the Yangtze River lies between the upstream and downstream regions. The spatial coordination of urban low carbon economy efficiency and digital finance in the Yangtze River Economic Belt region of China shows an increasing distribution from the upstream region to the downstream region. 

The above is our improvement and supplement to the “Spatial and temporal evolution of the CCD” section. “Spatial and temporal evolution of the CCD” section mainly answers the second question raised in the introduction section: how has the coupled coordination relationship between urban low-carbon economy efficiency and digital finance in China's Yangtze River Economic Belt changed over the past few years, and how will the spatial pattern change in the future?

(3)In the "Analysis of Influencing Factors of CCD ((Coupling coordination degree))" section, we add the characteristics of the changes of influencing factors such as population development level, informatization level, greening level, transportation level, industrial structure, and scientific research level in the influence level in 2012-2019 from a general perspective. Overall, the level of population development, level of informatization, level of greening, level of transportation, industrial structure, and level of scientific research are all drivers of the coupled coordination of ULEE and digital finance in the YEB region of China. Among them, the influence of informatization level and industrial structure on the coupling coordination tends to increase gradually, while the influence of population development level, greening level, transportation level, and scientific research level on the coupling coordination gradually decreases. In addition, we have added more quantitative data to the analysis of specific influencing factors. The influence of population development level is slowly decreasing in the overall trend, with an average annual decrease of 0.35%. However, the impact of this factor on coupling coordination is still at a relatively high level compared with other factors, and the impact of this factor on coupling coordination gradually increases in 2012-2014, reaching the maximum in 2014, and then tends to decrease in 2015-2019. The impact of the level of informatization shows an overall increasing trend with an average annual increase of 1.17%. the impact of this factor on the coupling coordination is the largest in 2014 and the smallest in 2016. The influence of greening level shows a slow decreasing trend overall, with an average annual decrease of 0.33%. The influence showed an increasing trend from 2012-2015, after which the influence of this factor gradually decreased. The impact of traffic and transportation level as a whole shows a slow decreasing trend with an average annual decrease of 0.29%, and the impact gradually increases from 2012-2017, after which the influence of this factor decreases significantly in a trend. The impact of industrial structure shows an overall upward trend with an average annual growth of 0.73%, and the q-value of this factor reached the highest in 2016 and the lowest in 2018. The influence of scientific research level shows a decreasing trend, with an average annual decrease of 1.70% and q-values fluctuating between 0.3715-0.5989, and the q-value of this factor is still relatively high compared with other factors.

The above is our improvement and supplement to the "Analysis of Influencing Factors of CCD" section. "Analysis of Influencing Factors of CCD" section mainly answers the third question raised in the introduction section: what are the factors affecting the coordinated development of low-carbon economy efficiency and digital finance in the cities of China's Yangtze River Economic Zone?

3.We also remove some of the literature (Geng et al., 2020, doi: 10.1155/2020/8673965) in the “Results” section to avoid the problem of over-citation of literature.

Responses to the comments of reviewer #2

 We highly appreciate reviewer #2 for his insightful comments and criticism, which have helped us improve both the content and the presentation of our work.We have revised our manuscript, according to the reviewers’ comments and suggestions.

1.Recommended to add some quantitative analysis to the abstract.

Response:We appreciate the reviewers' comments, and we have made improvements to the “Abstract” section as suggested. We have added specific and quantitative analysis to the conclusions of the abstract, such as the average annual increase in coupling coordination of 3.42% and the spatial center of gravity of coupling coordination will shift 22.17 km to the southwest. In order to make our study suitable for wider audiences, we also explain the economic implications indicated by the increase in coupling coordination.

The “Abstract” section has been modified as shown below:

China is a large country with rapid economic expansion and high energy consumption, which implies that the country's overall carbon emissions are enormous. It is vital to increase urban low-carbon economy efficiency (ULEE) to achieve sustainable development of China's urban economy. Digital finance is a significant tool to boost ULEE by providing a convenient and effective funding channel for urban low-carbon economic transformation. Analyzing the coupled and coordinated relationship between ULEE and digital finance is of vital importance for the sustainable development of the urban economy. This paper selects panel data of 100 cities in China's Yangtze River Economic Belt (YEB) in 2011-2019, and analyzes the research methods such as the Global Malmquist-Luenberger index model, coupling coordination degree (CCD) model, standard deviation ellipse model, gray model, and geographic detector by The spatial and temporal distribution, dynamic evolution characteristics and influencing factors of the CCD between ULEE and digital finance are analyzed. The study shows that : (1) the CCD of ULEE and digital finance grows by 3.42% annually, reflecting the increasingly coordinated development of the two systems; (2) The CCD of ULEE and digital finance shows a distribution pattern of gradient increase from the upstream region of Yangtze River to the downstream region, meanwhile, the spatial center of gravity moves mainly in the midstream region; (3) The spatial center of gravity of CCD of ULEE and digital finance is expected to move 22.17 km to the southwest from 2019 to 2040; (4) In terms of influencing factors, the influence of informatization and industrial structure on the CCD increases over time, while the influence of factors such as population development, greening, transportation, and scientific research decreases over time. Finally, this paper proposes policy recommendations for improving the CCD of ULEE and digital finance based on the empirical results.

2.The contribution and innovation of manuscript can appropriately add some dialogues with the past literature.

Response: We thank reviewer #2 for his suggestion, which we have followed to improve and add to the contributions and innovations in the article. We have added interactions with past literature in the presentation of the contributions and innovations of the article to highlight the innovative nature of the article. We can find after reviewing the study of Liu (2019) that the current research mainly focuses on measuring and analyzing the regional low carbon economy efficiency itself (doi: 10.15666/aeer/1703_64296444), and there is a gap in the research on the coupled and coordinated relationship between urban low carbon economy efficiency and digital finance. Meanwhile, a review of Meng's (2018) study shows that the existing low-carbon economy efficiency evaluation index system considers a single perspective and needs to be improved (doi: 10.1016/j.jclepro.2018.07.219), ignoring the role of public finance and land use in the development of low-carbon economy in cities. Therefore, this paper constructs a new evaluation system of urban low-carbon economic efficiency from the perspective of public finance and land and investigates for the first time the coupled coordination relationship between low-carbon economy efficiency and digital finance in cities of the Yangtze River Economic Zone through a coupled coordination degree model is innovative. In addition, we have added three problems addressed by the article and specified the main contributions of the article.

After the revision, the article's elaboration in terms of contributions and innovations is shown as follows：

According to existing research, this paper finds that the current literature mainly focuses on calculating and analyzing the efficiency of low carbon economy in each region (Liu et al., 2019), while there is a gap in the research on the coupled coordination relationship between urban low-carbon economy efficiency (ULEE) and digital finance. Meanwhile, the existing literature lacks the analysis of the future spatial pattern of coupling coordination degree (CCD) between ULEE and digital finance and the exploration of the influencing factors of CCD. In addition, the assessment indicator system of low carbon economy efficiency in the existing literature considers a single perspective and needs to be improved (Meng et al., 2018). Therefore, to remedy these shortcomings, this paper focuses on addressing three issues regarding the developmental relationship between ULEE and digital finance in the Chinese YEB region:

·How has the ULEE in the YEB region of China changed over the past few years?

·How has the coupling and coordination relationship between ULEE and digital finance in China's YEB region changed over the past few years, and how will the spatial pattern change in the future?

·What are the factors affecting the coordinated development of ULEE and digital finance in the YEB region of China?

These questions are closely related to urban low-carbon transition as well as sustainable development. The most important contribution of this study is to provide the first reliable model for evaluating the level of coordinated development of ULEE and digital finance, using 100 cities in the YEB region of China as examples. Based on the standard deviation ellipse model, gray model, and geographic detector, the spatial and temporal evolution characteristics, future spatial patterns, and impact factors of the coupled and coordinated relationship between the ULEE system and digital financial system are analyzed for the first time. In addition, this study improves the evaluation framework of low carbon economy efficiency from the viewpoint of public finance and land usage. This research provides a new path to foster the coordinated development of low-carbon economy efficiency and digital finance in the YEB region, which helps the government to formulate environmental protection policies, accelerate the low-carbon urban transformation, and boost sustainable development.

3.Figures need to improve clarity.

Response: Thank reviewer #2 for his suggestion, we have adjusted the pictures. We increased the resolution of the image and appropriately increased the size of the text in the image, and finally checked the image with SPACE to ensure that the modified image meets PLOS ONE requirements. In addition, according to the suggestion of the journal, we deleted Figures 1 and 7 with maps in the original manuscript and used new figures instead of Figures 3, 6, and 9 with maps in the original manuscript.The adjusted images are as follows:

Figure 1. Trend of ULEE

Figure 2. Spatial distribution of ULEE

Figure 3. Trend of the CCD

Figure 4. Evolution of the CCD

Figure 5. Spatial distribution of the CCD

Figure 6. The changes of relevant parameters of SDE

Figure 7. Forecast of spatial pattern

4.Suggested to supplement the energy conversion factor value used in the calculation of carbon emissions.

Response: Thanks to reviewer #2 suggestion, we have added the energy conversion factors used to calculate carbon emissions to the revised article. The equation for accounting for urban carbon emissions can be expressed as follows:

In the above equation, denotes the carbon emission of the city in period , denotes the consumption of type of energy in the city in period , denotes the conversion factor of type of energy, the LPG conversion standard coal factor is 1.7143kgce/kg, artificial gas and natural gas conversion standard coal factor are 1.3300kgce/m3. denotes the carbon content factor of type of energy, the carbon content factor is 0.5041 kg/kgce for LPG and 0.4484 kg/kgce for manufactured gas and natural gas. denotes the carbon oxidation factor of type of energy, the carbon oxidation rate is 0.98 for LPG and 0.99 for manufactured gas and natural gas. denotes the amount ratio of carbon dioxide to carbon molecules (44/12), denotes the total annual electricity consumption of the city in period and denotes the baseline carbon emission factor for period in region where the city's grid is located.

5. Discussion is needed. By interacting with previous studies, the author should discuss more about the "story" behind the results and data.

Response: Thanks to the suggestion of REVIEWER #2, we have reviewed the relevant literature and have added and improved the “Results” section of the article. We have combed through the relevant literature and added to the analysis of why the results were generated. We have revised it in two parts:

1.In the “Analysis of urban low-carbon economy efficiency (ULEE)” section, we add the reasons for the changes in the temporal characteristics of ULEE. The ULEE of China's Yangtze River Economic Belt (YEB) showed an overall fluctuating upward trend from 2012 to 2019, with an average annual growth of 0.93%, which is mainly due to the fact that since 2012, the Chinese government has formulated a series of policies to promote the development of low-carbon economy, encouraging enterprises to use clean energy and improve energy use efficiency, so that the carbon reduction technology in the cities of YEB region is widely and deeply utilized (Lin et al., 2022, doi: 10.1016/j.apenergy.2021.118160). In 2015, ULEE increased by 1.4%, technological advancement changes (TC) increased by 7.84%, while PEC and SEC decreased by 1.56% and 3.66%, respectively, which shows that the growth of ULEE in 2015 was mainly driven by the growth of TC. TC experienced high growth in 2015 mainly because the Chinese government strengthened its financial support for low-carbon industries and further implemented tax relief policies in 2015, which promoted low-carbon technology innovation among enterprises (Hadfield et al., 2019, doi: 10.1080/08111146.2017.1421532). In addition, the government increased the elimination of backward production capacity, adjusted and optimized the urban energy consumption structure, reduced the proportion of coal consumption, and encouraged the development of new clean energy, thus accelerating the development of low-carbon economy. After 2017, ULEE started to show a sequential growth, increasing by 4.41% and 4.07% in 2018 and 2019, respectively. This may be due to the official launch of the carbon emission trading market at the end of 2017. Under the market mechanism, market players with excess or insufficient allowances can accomplish their emission reduction targets more effectively through trading. The carbon trading mechanism reduces the cost of carbon reduction for enterprises, accelerates low-carbon technology innovation and diffusion (Lyu et al., 2020, doi: 10.1080/17583004.2020.1721977), and at the same time promotes the coordinated development of industrial structure, thus achieving efficiency improvements in a low-carbon economy. 

We also analyze the overall spatial distribution characteristics of ULEE in the upper, middle and lower reaches of the Yangtze River using indicator decomposition, and discuss the reasons for the formation of the distribution and the inspiration given by the distribution. In general, the ULEE in the YEB region of China shows a spatial distribution of the strongest in the upstream region, the second strongest in the downstream region, and the weakest in the midstream region, and the mean values of ULEE in the upstream, downstream, and midstream regions are 1.0172, 1.0077, and 1.0062, respectively. after decomposing the ULEE, it is found that the decomposed indicators show different characteristics of spatial distribution in the YEB region, respectively. The spatial distribution of pure technical efficiency (PEC) is similar to that of ULEE, with the highest PEC level in the upstream region at 1.0198 and the lowest PEC level in the midstream region at 1.0118, while the PEC level in the downstream region is between the upstream and midstream regions at 1.0155. The spatial distribution of scale efficiency (SEC) in the YEB region is mainly characterized by decreasing from the upstream region to the downstream region, with 1.0110, 1.0050, and 1.0030 for the upstream, middle, and downstream regions, respectively. The spatial distribution of TC in the YEB region is characterized by the strongest in the downstream region, followed by the upstream region, and the weakest in the midstream region. The TC in the downstream, upstream and midstream regions were 0.9978, 0.9946 and 0.9934, respectively. From the spatial distribution of the decomposition indicators, it can be seen that the ULEE of the upstream region is relatively the highest mainly due to the efficient application of technology and scale dividend, while the region needs to improve in terms of technological innovation. The low-carbon economy in the midstream region is at the stage of scale development, and should expand production investment and accelerate technological innovation to further improve ULEE, while the PEC level in this region is the lowest among the three regions, and this region should strengthen the application and promotion of low-carbon technologies. The application level of low-carbon technologies in the downstream region is relatively high, and the region outperforms other regions in terms of technological innovation, but the strength of technological R&D is still insufficient. Meanwhile, the SEC level in the downstream region is lower than that in the upstream and midstream regions, and the scale dividend generated by expanding resource input in this region is relatively the least.

2.In the “Spatial and temporal evolution of the coupling coordination degree (CCD)” section, we add the reasons for the changes in the temporal characteristics of CCD of ULEE and digital finance in 2012-2019. The CCD has been growing annually from 2012 to 2016, from 0.44 to 0.65. However, the value of CCD is not high and the growth rate has been decreasing year by year. This is mainly because digital finance is still in the early stage of development and the intensity of support for the development of low-carbon economy is insufficient, which leads to the slow growth of ULEE and thus slows down the coordinated development of ULEE and digital finance (Zhao et al., 2021 ,doi: 10.3390/su132112303). CCD showed a slight decline in 2017 to 0.64, which is related to the decline of ULEE due to insufficient technological innovation, which limits the synergistic development of ULEE and digital finance. After 2017, CCD has showed continuous growth, increasing to 0.71 in 2019. This is due to the fact that the launch of China's carbon emission rights market has enabled effective control of urban carbon emissions, while the opening of the carbon emission rights market has stimulated industrial structure optimization and energy consumption restructuring (Tan et al., 2022, doi:10.1016/j.apenergy.2022.118583), accelerating the development of low-carbon industries, which is conducive to promoting the development of the digital finance industry. And the development of digital finance provides more financing opportunities for urban low-carbon economic transformation.

We also add the reasons for the formation of the spatial distribution characteristics of CCD in China's Yangtze River Economic Belt at different time points. the overall level of coupling coordination of cities in the Yangtze River Economic Belt was relatively low in 2012. The cities with approaching imbalance (0.40<CCD<0.49) and slight imbalance (0.30<CCD<0.39) are mainly concentrated in the upstream region of the Yangtze River, and the cities with reluctant coordination (0.50<CCD<0.59) are mainly concentrated in the midstream and downstream regions of the Yangtze River. This spatial distribution characteristic may be due to the fact that cities in the upstream region are mostly inland cities, where labor, technology, capital, and other resources are difficult to obtain, and production and operation are mostly based on rough industrial production, making it difficult to coordinate the development of urban economy and urban environment (Fang et al., 2021, doi: 10.1016/j.ecolind.2021.107864). Meanwhile, the development of the digital finance industry in the region is relatively slow and the financing support is insufficient, and the low-carbon transformation of cities is constrained by the sluggish development of digital finance, forcing the two to achieve coordination with increased difficulty. Compared with 2012, the overall level of coupling coordination of cities in the Yangtze River Economic Belt improved in 2014, except for Chengdu, Zunyi and Zhaotong, which are located in the upper Yangtze River region and are still in a state of approaching imbalance, most of the cities in approaching imbalance transformed to reluctant coordination or slight coordination, mainly concentrated in the middle and lower reaches of the Yangtze River region. This is probably because the middle and lower reaches of the Yangtze River have developed economies, advanced energy-saving and emission reduction technologies, reasonable energy consumption structures and relatively high levels of digital finance development, and digital finance can play a better financing role in the development of low-carbon economies in cities, so the development relationship between ULEE and digital finance in the middle and lower reaches of the Yangtze River is more coordinated. In 2016, the coupling coordination of cities in the Yangtze River Economic Belt further improved, and some slightly coordinated cities transformed to intermediate coordination (0.70<CCD<0.79), such as Suzhou, Zhenjiang, Quzhou and Nanchang, mainly concentrated in the middle and lower reaches of the Yangtze River region. The reluctantly coordinated cities are mainly distributed in the upper Yangtze River region, while the slightly coordinated cities show an even distribution throughout the Yangtze River Economic Belt, with a denser distribution in the middle and lower Yangtze River region. This may be due to the fact that as the development level of digital finance in the YEB region improves, the financing channels for each city to carry out low-carbon transformation are expanded, low-carbon technology R&D and industrial restructuring are better supported, and the coupled coordination relationship between digital finance and ULEE is optimized.

6. Originality and novelty of the paper needs to be further improved and clarified.

Response: Thanks to the advices of REVIEWER #2, we have clarified and improved the originality and novelty aspects of the article. After reviewing the literature, we have added in the “Literature review” section the current knowledge gaps in the researches on the coupled and coordinated relationship between ULEE and digital finance, and how our research fills these gaps. The contents are as follows：

A review of the literature reveals that there are two specific knowledge gaps regarding the developing relationship between ULEE and digital finance.

·The current low carbon economy efficiency assessment index system is constructed from a single perspective and needs to be improved.

For most studies on measuring low-carbon economy efficiency, the assessment framework is mainly constructed around the capital, labor, energy, GDP, and carbon emissions (Zhang et al., 2017), ignoring the role of public finance and land use in the advance of the low-carbon economy. Public finance can effectively stimulate low-carbon innovation (Owen et al., 2018), which has an important role in the low-carbon transition of cities. Meanwhile, land use is the second-largest source of carbon emissions after fossil fuels (Huang et al., 2015). Considering the importance of public finance and land use to the growth of the urban low-carbon economy, this paper includes them in the evaluation index system of ULEE.

·Till now, there is no suitable indicator to measure the change in the coupled and coordinated relationship between ULEE and digital finance.

Traditional finance has undergone a digital transition, and digital finance is a new manifestation of that development. Digital finance can provide users with more convenient and efficient financial services with the help of the Internet and digital technology, and significantly increase the efficiency of financing for individuals and enterprises (Hu et al., 2016). The existing researches have mainly concentrated on analyzing the relationship between low-carbon economy and traditional finance (Zhang et al., 2011), and there are gaps in the research on the relationship between ULEE and digital finance, especially the research on the coupling and coordination relationship between the two systems. The coupling and coordination relationship between ULEE and digital finance lacks theoretical mechanisms and suitable measurement indicators. In addition, the influencing factors and future spatial pattern prediction of the CCD between the two systems also need to be studied. The study of the CCD between ULEE and digital finance and the influencing factors will help the coordinated development of the low-carbon economy and digital finance, accelerate the low-carbon transformation of Chinese cities, promote the process of global CO2 emission reduction, and enhance sustainable development.

Accordingly, this study endeavors to narrow these gaps through the following efforts.

·Improved the assessment indicator system of ULEE from the perspective of public finance and land use.

This research proposes a new evaluation index system of ULEE from seven aspects: capital, labor, public finance, land, energy, GDP, and carbon emission, and measures the ULEE of 100 cities in China's YEB region using the GML index model. Meanwhile, this paper analyzed the changes of ULEE from the time perspective, analyzed the distribution differences of ULEE from the spatial perspective, and decomposed the ULEE index to explore the reasons for the changes in ULEE.

·Measured the CCD of ULEE and digital finance in 100 cities in the YEB region of China from 2012 to 2019, and analyzed its influencing factors.

This paper measures the coupled coordination relationship between ULEE and digital finance in the YEB region for the first time by combining ULEE and Peking University Digital Inclusive Finance Index using the CCD model and conducts a spatio-temporal evolution analysis with the SDE model to discuss the changes in the CCD of the two systems and the reasons. Meanwhile, this paper also conducts a time series prediction of the parameters of SDE based on the gray model and analyzes for the first time the changes in the spatial distribution of the CCD of ULEE and digital finance in 2019-2040. In addition, to further promote the coordinated development of ULEE and digital finance and sustainable urban development, this paper detects for the first time the changes in the influence of informatization, industrial structure, population development, greening, transportation, and scientific research on CCD between ULEE and digital finance by using geographic detectors.

7. Writing needs to be significantly improved.

Response: Thanks to reviewer #2 for his comments. Considering that we need to improve our writing, we adjusted the language and improved the “Introduction” section, the “Literature review” section, the “Results” section, and the “Conclusion” section of the article, and sent the manuscript to a foreign institution for language touch-ups. We believe that the revised article will be improved in terms of writing.

Thank you again for your positive and constructive comments and suggestions on our manuscript. We have tried our best to improve the manuscript and made some changes in the manuscript. These changes will not influence the framework of paper. 

We appreciate for Editors and Reviewers’ warm work earnestly, and hope the correction will meet with approval.

yours sincerely,

Fengge Yao, Liqing Xue, Jiayuan Liang 

Corresponding author:

Name: Jiayuan Liang 

E-mail:timljy1994@gmail.com

---

## [Decision Letter · Decision Letter 1]

2 Jun 2022

PONE-D-22-00910R1Research on Coupling Coordination and Influencing Factors between Urban Low-carbon Economy Efficiency and Digital Finance – Evidence from 100 Cities in China’s Yangtze River Economic BeltPLOS ONE

Dear Dr. Liang,

Thank you for submitting your manuscript to PLOS ONE. After careful consideration, we feel that it has merit but does not fully meet PLOS ONE’s publication criteria as it currently stands. Therefore, we invite you to submit a revised version of the manuscript that addresses the points raised during the review process.

We look forward to receiving your revised manuscript.

Kind regards,

Xingwei Li, Ph.D.

Academic Editor

PLOS ONE

Journal Requirements:

Additional Editor Comments:

The reviewers have provided comments for further revision of this manuscript, please revise in accordance with these comments.

Reviewers' comments:

Reviewer's Responses to Questions

**Comments to the Author**

1. If the authors have adequately addressed your comments raised in a previous round of review and you feel that this manuscript is now acceptable for publication, you may indicate that here to bypass the “Comments to the Author” section, enter your conflict of interest statement in the “Confidential to Editor” section, and submit your "Accept" recommendation.

Reviewer #1: (No Response)

Reviewer #2: All comments have been addressed

2. Is the manuscript technically sound, and do the data support the conclusions?

Reviewer #1: (No Response)

Reviewer #2: Yes

3. Has the statistical analysis been performed appropriately and rigorously? 

Reviewer #1: (No Response)

Reviewer #2: Yes

4. Have the authors made all data underlying the findings in their manuscript fully available?

Reviewer #1: (No Response)

Reviewer #2: Yes

5. Is the manuscript presented in an intelligible fashion and written in standard English?

Reviewer #1: (No Response)

Reviewer #2: Yes

6. Review Comments to the Author

Reviewer #1: Accept as it is

Accept as it is

Accept as it is

Accept as it is

Accept as it is

Accept as it is

Accept as it is

Reviewer #2: 1. Figures need to improve clarity.

2. It is recommended that the Discussion section be a separate chapter.

7. PLOS authors have the option to publish the peer review history of their article (what does this mean?). If published, this will include your full peer review and any attached files.

Reviewer #1: No

Reviewer #2: No

---

## [Author Response · Author response to Decision Letter 1]

30 Jun 2022

Dear Editors and Reviewers,

Thank you very much for your kindly comments concerning our manuscript entitled “Research on Coupling Coordination and Influencing Factors between Urban Low-carbon Economy Efficiency and Digital Finance -- Evidence from 100 Cities in China's Yangtze River Economic Belt.” (PONE-D-22-00910R1). Your comments and those of the reviewers are all valuable and very helpful for revising and improving our paper, as well as the important guiding significance to our researches. We have studied comments carefully and have made correction which we hope meet with approval. Revisions in the text are shown using yellow highlight [example] for additions, and strikethrough font [example] for deletions. We hope that the revisions in the manuscript and our accompanying responses will be sufficient to make our manuscript suitable for publication in PLOS ONE. The main corrections in the paper and the responds to the reviewer’s comments are as flowing.

Responds to the Journal’s comments:

Response: We fully respect the requirements of the journal. We have checked the reference list for the latest revised manuscript and confirmed that it is complete and correct. We have not cited a retracted article as a reference. We have made the following changes to the reference list, we have added a new reference (DOI: 10.3389/fenvs.2022.886886), added page numbers to some references (DOI: 10.1016/j.techfore.2020.120485; DOI: 10.3390/ buildings12010083), and the DOI of one reference was corrected (DOI: 10.15666/aeer/1705_1224512258).

Responds to the additional editor’s comments:

1. The reviewers have provided comments for further revision of this manuscript, please revise in accordance with these comments.

Reponse: We humbly accepted the additional editor’s comments, fully considered the comments of reviewer #2. In accordance with the reviewers’ suggestion, we have made the following main changes to the article:

1. We have improved the clarity of the images. We increased the resolution of the images and checked them using the PACE tool recommended by the journal to ensure they meet the journal's requirements.

2. We have revised the discussion section of the article into a separate section as suggested by the reviewer.

Responds to the reviewer’s comments:

Responses to the comments of reviewer #1

1. Accept as it is

Accept as it is

Accept as it is

Accept as it is

Accept as it is

Accept as it is

Accept as it is

Response: We are very grateful to reviewer #1 for his review suggestions, which have been very helpful in improving the article.

Responses to the comments of reviewer #2

 We highly appreciate reviewer #2 for his insightful comments and criticism, which have helped us improve both the content and the presentation of our work.We have revised our manuscript, according to the reviewers’ comments and suggestions.

1. Figures need to improve clarity.

Response:We appreciate the reviewers' comments, and we have improved the clarity of the images. We upgraded the resolution of each image and checked the images using the journal's recommended PACE tool to ensure they met the journal's requirements. We have found that viewing the images directly through the PDF file generated by the submission system may not allow you to see the images clearly because they are compressed. We recommend that reviewer #2 download the images by clicking on the link at the top right of the images in the PDF and then view them, as this will ensure that the images are clear. Adjusted image, as “Response to Reviewers.docx”. 

2. It is recommended that the Discussion section be a separate chapter.

Response: We are very grateful for the comments made by reviewer #2 and we have revised the Discussion section of the paper into a separate section. In the Disscusion section, we explain the findings of the article and answer the research questions posed in the Introduction section. We also further elaborate on the significance of this study in relation to previous research and explain the limitations of this study in Disscusion section. In addition, we have added a reference in Disscusion section (DOI: 10.3389/fenvs.2022.886886).

The Discussion section of the revised manuscript reads as follows:

This paper measured the ULEE of 100 cities in the YEB region of China and then analyzed the spatial and temporal evolutionary trends of the coupled coordination of ULEE and digital finance and the changes in the main influencing factors in conjunction with the Peking University Digital Inclusive Finance Index. These findings can help policy research departments to understand the coordinated development relationship between ULEE and digital finance in the YEB region of China in recent years, to better promote the two systems to achieve quality and harmonious development.

First, from a time evolution perspective, ULEE showed an overall fluctuating growth trend from 2012-2019, with a total increase of 7.41%. On the one hand, PEC and SEC grown by 13.16% and 4.8% respectively, indicating that the growth of ULEE is mainly driven by PEC and SEC. This is because the Chinese government has formulated a set of policies to boost the growth of the low-carbon economy since 2012 to encourage enterprises to use clean energy and improve energy use efficiency, which has led to the widespread and in-depth use of carbon reduction technologies in cities in the YEB region (Lin et al., 2021). On the other hand, the TC declined, which also indicates that the lack of science and technology innovation in the YEB region has limited the further progress of the efficiency of low-carbon economic development in the region. However, the TC has shown a significant increase in 2015, which may be attributed to the Chinese government's enhanced financial support and further implementation of tax breaks for low-carbon industries in 2015, which has boosted low-carbon technological innovation in enterprises (Hadfield et al., 2019). Therefore, promoting low-carbon production technology innovation in the R&D sector and developing new clean energy sources will be an important direction for the growth of the low-carbon economy in cities in the YEB region in the future. In addition, ULEE started to show continuous and stable growth after 2017, which may be related to the official launch of the carbon emission trading market at the end of 2017. Under the market mechanism, market players with excess or insufficient allowances can meet their emission reduction targets more effectively through trading. In other words, the carbon trading mechanism reduces the cost of carbon reduction for enterprises and accelerates the innovation and diffusion of low-carbon technologies (Lyu et al., 2020), thus promoting the optimization of industrial structure and realizing the efficiency of the low-carbon economy. From the perspective of spatial distribution, the upstream region had the highest ULEE, which mainly benefited from the efficient application of low-carbon technologies and the spillover of scale dividends, but there is still room for improvement in technological innovation in this region. The midstream region had the lowest PEC among the three regions, and its low-carbon economy was at the scale development stage. Therefore, the midstream region should accelerate low-carbon technology innovation and strengthen the application and diffusion of low-carbon technologies. The downstream region outperformed the other regions in terms of low carbon technology innovation, but had the lowest level of SEC, indicating that the downstream region had the relatively least scale dividend. From the perspective of city characteristics, Yuxi, Jingzhou, Zunyi, Lijiang, Qujing, Guangyuan, Changzhou, and Shanghai had high levels of ULEE among the YEB regions in China. Among them, Yuxi, Jingzhou, Zunyi, Lijiang, Qujing, and Guangyuan, inspired by China's high-quality development goals, have rapidly developed low-carbon tourism and significantly increased their fiscal revenues, thus implementing a set of policies to boost energy restructuring, including rational use of natural superiorities to exploit low-carbon energy, improve energy use efficiency and vigorously promote low-carbon lifestyles to the public, which have greatly enhanced the cities' ULEE. As for Changzhou and Shanghai, both cities have relatively sound infrastructure, reasonable industrial structures, convenient transportation, a concentration of talents, and a strong capacity for low-carbon technological innovation. Shanghai, in particular, as the world's financial center, has a high level of digital financial development, which offers efficient financial support for the development of green technologies and is contributive to the study, development, and promotion of new clean energy sources, which can significantly improve the city's ability to reduce carbon emissions. The ULEE of Zhaotong, Taizhou, Ziyang, Huzhou, Jiaxing, Dazhou, Changsha, Fuzhou, Nanchang, Yichang, Wenzhou, Liu'an, Loudi, and Jingmen was at a relatively low level in the YEB zone, primarily due to the low level of TC in these cities, and therefore the innovation and application of low-carbon technologies should be strengthened. The low PEC in Xianning and Yichun restricted the improvement of ULEE, so optimizing the application of technology and improving the management of technology should be the focus of future development in these cities. The mean values of SEC in Ji'an and Jingmen were less than 1. They need to expand their production scale and increase factor inputs for low-carbon economic development, to improve ULEE. In summary, this paper here discussed the spatio-temporal evolution characteristics of ULEE in China's YEB region from 2012-2019, answering the first question raised in Section 1.

Second, from the perspective of time evolution, CCD showed an overall trend of stable growth from 2012-2019. Specifically, the CCD grew slowly from 2012-2016. This is mainly because digital finance was still in its infancy and the intensity of support for the growth of the low-carbon economy was insufficient, resulting in slow growth of ULEE and thus slowing down the coordinated development of ULEE and digital finance (Zhao et al., 2021). CCD has continued to grow after 2017, probably due to the rapid growth of China's digital finance sector and the launch of China's carbon emission rights market, which have stimulated the advance of low-carbon industries and the restructuring of energy consumption, promoting urban carbon reduction (Tan et al., 2022), thus allowing for a higher level of coordination between the two systems. From the perspective of spatial evolution, most of the cities in the YEB experienced a shift from approaching imbalance and slight imbalance to primary coordination and intermediate coordination during the study period, which indicates that the coordinated development relationship between ULEE and digital finance in the YEB region has been widely improved. Among them, the lower Yangtze River region had the highest degree of digital finance development relative to other regions, allowing cities to receive more abundant and efficient financial support in the process of low-carbon economic development, thus promoting clean technology research and development and improving the energy consumption structure to achieve a high level of CCD development. The cities in the upstream region are mostly inland cities, where access to resources such as labor, technology, and capital is difficult, and production and operation are mostly based on crude industrial production, making it difficult for the urban economy and urban environment to develop in a coordinated manner (Fang et al., 2021). Meanwhile, the growth of digital finance in the region is comparatively lagging, with insufficient financing support, making it hard for the two to accomplish coordinated evolution, and therefore the CCD level in the region was the lowest. The value of the CCD in the middle reaches of the Yangtze River lies between the CCD in the upper reaches and those in the lower reaches. In addition, the cities in the YEB region with a relatively high level of coordinated development of ULEE and digital finance are mainly located in the northeast-southwest direction. The spatial center of gravity shifted first to the northeast, then to the northwest, and finally to the southwest during the study period. This is mainly because the CCDs of cities such as Liu'an, Zhenjiang, and Yancheng, which are located in the northeast direction, showed a significant increase from 2012-2014. CCDs in cities such as Mianyang and Chengdu, located in the northwest direction, increased significantly from 2014-2016. CCDs in cities in the southwest of the YEB increased at a faster rate after 2016, while CCDs in cities in the northeast maintained a relatively low rate of growth. Thus, a trajectory of the center of gravity moving first to the northeast, then to the northwest, and finally to the southwest was formed. In terms of the future direction of the spatial pattern, the spatial center of gravity of CCD will move southwestward in the middle region of the YEB in 2019-2040, indicating that the western cities in the middle Yangtze River region are expected to boost the coordinated growth of ULEE and digital finance in the future. From 2019 to 2040, the discrete level of CCD in the northeast-southwest direction will show a downward trend, while the discrete level in the northwest-southeast direction will increase, indicating that cities in the northwest-southeast direction have the possibility of rapidly increasing CCD, and the exchange of low-carbon technologies, financial resources, digital technologies and technical talents between regions will be further strengthened, thus benefiting cities with relatively low CCD levels. In addition, the spatial pattern of CCD in 2019-2040 is dominated by the northeast-southwest direction and will turn in a counterclockwise direction, again indicating that CCD in cities with a northwest-southeast direction has a strong potential to improve in the future. In summary, this paper here discusses the spatio-temporal evolutionary characteristics of the CCD of ULEE and digital finance from 2012-2019 and the future spatial pattern changes, answering the second question posed in Section 1.

Third, in terms of influencing factors, the level of informatization and industrial structure tended to gradually increase their influence on CCD, while the level of population development, greening, transportation, and scientific research gradually weaken their influence on CCD. In terms of the influence of the level of population development, the influence of population development showed a slow downward trend overall, with an average annual decrease of 0.35%. However, the influence of this factor was still at a relatively high level compared to other factors. This is mainly because as the number of people increases, there is a corresponding increase in social labor resources and consumption behavior of the population, which helps to boost the efficiency of economic development while promoting the development of the digital financial sector. Changes in low-carbon consumption behavior would also have a vital influence on carbon emissions. As a result, the influence of the level of population development on CCD was slowly diminishing but still at a high level. In terms of the impact of the level of informatization, the impact of the level of informatization showed an overall trend of growth over time, with an average annual growth of 1.17%. This is because the popularity of the Internet is the basis for the creation and advance of digital finance, and digital financial services need to be realized through the Internet and related information technology. And the construction of information technology has given impetus to the birth of more new green industries, providing new impetus to the growth of the low-carbon economy. In terms of the impact of greening levels, the overall impact of greening levels showed a slow downward trend, with an average annual decrease of 0.33%. This may be because with the deepening of greening construction, the space available for greening construction in built-up areas gradually becomes less and the marginal contribution of greening level to the low-carbon economy gradually decreases, and the driving effect of greening construction on the low-carbon economy diminishes, thus obstructing the coordinated advance of ULEE and digital finance. In terms of the influence of transport levels, the influence of this influence showed an overall decreasing trend, with the influence increasing year on year from 2012-2017, but decreasing sharply after 2017, which should be attributed to the increase in the number of public-operated trams from 2012-2017, the increase in the number of low-carbon trips made by urban residents and the corresponding increase in urban transport capacity, which led to a good development of urban low-carbon economic efficiency, but in recent years the number of public-operated trams in the city gradually saturated and the number of private cars in the city increased sharply, boosting a raise in CO2 emissions. In terms of the influence of industrial structure, the influence of this factor on CCD increased gradually, indicating that the Chinese government issued relevant policies to promote the proportion of tertiary industries in GDP, eliminating industries related to high pollution, high energy consumption, and high carbon emissions, and services such as digital finance have been better developed, and further improvement of industry construction is an important direction to coordinate the development of ULEE and digital finance. In terms of the influence of the level of scientific research, the influence of the level of scientific research on CCD decreased overall, but its influence was still at a high level compared to other factors, which indicates that improving the level of scientific research and innovation is still a priority for the development of cities in the YEB region of China and that cities should actively accelerate the upgrading of energy conservation and digital information technology, while ensuring that cities make effective use and promotion of existing technology, fully applying advanced technology to the growth of low carbon economies in cities, and accelerating the construction of digital financial services so that digital finance can provide strong financial support for the growth of the low carbon economy in the future. In summary, this paper discussed here the impact of factors such as level of population development, level of informatization, level of greening, level of transportation, industrial structure, and level of scientific research on the CCD of ULEE and digital finance in the YEB region of China from 2012-2019, answering the third question posed in Section 1.

Besides the above answers to the research questions, this paper provided several novel contributions to related studies. Firstly, in terms of ULEE evaluation, this paper complemented the framework for evaluating low-carbon economic efficiency in previous studies (Meng et al., 2018; Zhang et al., 2017). Compared with existing researches, this paper improved the evaluation framework of low carbon economy efficiency in terms of land use and financial support. Other studies on low-carbon economy efficiency can be further expanded based on this, such as the interaction between low-carbon economy efficiency and the level of technological development (Yang et al., 2022). Secondly, this study provided a new quantitative analysis perspective and is the first study to analyze the changes in the level of coordinated development of ULEE and digital finance through a coupled coordination degree model, which helps to help relevant sectors understand the coordinated development relationship between ULEE and digital finance. Thirdly, this paper further analyzed the expected changes in the spatial pattern of CCD between ULEE and digital finance and the influencing factors, which will provide a basis for local governments in China to formulate policies to enhance the level of coordinated development of ULEE and digital finance.

In addition, there are still some limitations to this paper. Firstly, the paper uses the latest data that can be collected at present, but there is still a certain lag, for further research can be done after the data is updated. Secondly, this paper does not take into account the climate change factor in the ULEE evaluation framework, and the model can be expanded in future research to improve the ULEE evaluation framework. Thirdly, this paper only considers the effect of individual factors on the level of coordinated growth of ULEE and digital finance but does not analyze the interaction of multiple influencing factors, which can be discussed in more depth in future research.

Thank you again for your positive and constructive comments and suggestions on our manuscript. We have tried our best to improve the manuscript and made some changes in the manuscript. These changes will not influence the framework of paper. 

We appreciate for Editors and Reviewers’ warm work earnestly, and hope the correction will meet with approval.

yours sincerely,

Fengge Yao, Liqing Xue, Jiayuan Liang 

Corresponding author:

Name: Jiayuan Liang 

E-mail:timljy1994@gmail.com

---

## [Editor Report · Decision Letter 2]

1 Jul 2022

Research on Coupling Coordination and Influencing Factors between Urban Low-carbon Economy Efficiency and Digital Finance – Evidence from 100 Cities in China’s Yangtze River Economic Belt

PONE-D-22-00910R2

Dear Dr. Liang,

We’re pleased to inform you that your manuscript has been judged scientifically suitable for publication and will be formally accepted for publication once it meets all outstanding technical requirements.

Kind regards,

Xingwei Li, Ph.D.

Academic Editor

PLOS ONE

Additional Editor Comments (optional):

The authors have carefully revised their manuscript and the current version is acceptable.
---

## [Editor Report · Acceptance letter]

7 Jul 2022

PONE-D-22-00910R2 

Research on Coupling Coordination and Influencing Factors between Urban Low-carbon Economy Efficiency and Digital Finance – Evidence from 100 Cities in China’s Yangtze River Economic Belt 

Dear Dr. Liang:

I'm pleased to inform you that your manuscript has been deemed suitable for publication in PLOS ONE. Congratulations! Your manuscript is now with our production department. 

Kind regards, 

on behalf of

Prof. Dr. Xingwei Li 

Academic Editor

PLOS ONE